# TQA-Bench: Evaluating LLMs for Multi-Table Question Answering

## Abstract

The advance of large language models (LLMs) has unlocked great opportunities in complex data management tasks, particularly in question answering (QA) over complicated multi-table relational data. Despite significant progress, systematically evaluating LLMs on multi-table QA remains a critical challenge due to the inherent complexity of analyzing heterogeneous table structures and the potentially large scale of serialized tabular data. Existing benchmarks primarily focus on single-table QA, failing to capture the intricacies of connections across multiple relational tables, as required in real-world domains such as finance, healthcare, and e-commerce. To bridge this gap, we present TQA-Bench, a new multi-table QA benchmark designed to evaluate the capabilities of LLMs in tackling complex QA tasks over complicated relational data. Our benchmark incorporates diverse relational database instances sourced from real-world public datasets and introduces a flexible sampling mechanism to create tasks with varying multi-table context lengths, ranging from 8K to 64K tokens. To further ensure robustness and reliability, we integrate symbolic extensions into the evaluation framework, enabling the assessment of LLM reasoning capabilities beyond simple data retrieval or probabilistic pattern matching. We systematically evaluate a range of LLMs, both closed-source and open-source, spanning model scales from 2 billion to 671 billion parameters. Our extensive experiments reveal critical insights into the performance of LLMs in multi-table QA, highlighting both challenges and opportunities for advancing their application in complex, data-driven environments.

## 1 Introduction

The rise of large language models (LLMs) (Jin et al., 2022) has unlocked unprecedented opportunities for tackling complex data management tasks (Biswal et al., 2024; Chen et al., 2024; Patel et al., 2024; Wornow et al., 2024), particularly in question answering (QA) across intricate relational data (Chen et al., 2020b; Gu et al., 2022; Pal et al., 2023; Zhang et al., 2024b; Zhu et al., 2024; He et al., 2024). Despite these advancements, systematically evaluating LLMs on multi-table QA remains a significant challenge due to the task's inherent complexity - multi-table QA requires LLMs to extract and analyze information from multiple interconnected tables, often dealing with highly heterogeneous table structures and serialized lengths. As LLMs continue to demonstrate remarkable capabilities across various data management applications (Patel et al., 2024; Wornow et al., 2024; Madden et al., 2024; Jiang et al., 2024), we believe there is an urgent need for *a comprehensive understanding of LLMs' performance in tackling the complexities of multi-table QA*.

Systematically evaluating and understanding the performance of LLMs on multi-table QA is a crucial step toward unlocking their full potential for data management and business intelligence (Chen et al., 2020b; Lei et al., 2023; Wu et al., 2025b). Structured relational data is pervasive across domains such as finance (Zhu et al., 2021), healthcare (Zhu et al., 2019), and e-commerce (Gao et al., 2021). Real-world tasks often require the processing, retrieving, and analyzing of multiple tables to support data-driven decision making. However, there is *a significant gap* between the existing Table QA benchmarks (Chen et al., 2020b; Lei et al., 2023; Wu et al., 2025b) and the practical demands of applications that operate on real-world tabular data. We believe that addressing this disparity is essential to bridge the divide and advance the utility of LLMs in complex, data-centric environments.

We list the current table QA benchmarks in Appendix §A (Table 7), and summarize the challenges of constructing a practical table QA benchmark from three key aspects. **First**, most existing Table

QA benchmarks are designed based on single-table contexts (Pasupat & Liang, 2015; Iyyer et al., 2017; Nan et al., 2022; Chen et al., 2020c;a; 2021; Katsis et al., 2021; Cheng et al., 2021; Wu et al., 2025b; Zhu et al., 2021), which fail to capture complex relationships across multiple interconnected tables in real-world scenarios. **Second**, the tables included in these benchmarks are often limited to tables with tiny data volumes, which do not reflect the most advanced LLMs' abilities since they can process millions of tokens (Dubey et al., 2024a; OpenAI, 2024a). **Third**, evaluating analytical abilities over a fixed set of tables and questions raises concerns regarding the reliability and generalizability of the results, as models may merely learn to exploit probabilistic patterns - that is, relying on dataset artifacts (answer-frequency priors, spurious word overlaps, recurring operator templates) instead of genuine cross-table reasoning - in the dataset rather than exhibit robust performance on genuinely complex multi-table queries (Mirzadeh et al., 2024).

To address these challenges, we propose a novel design for multi-table QA benchmarks. **First**, going beyond the simplistic single-table setups commonly used in current benchmarks, we construct the benchmark by collecting multi-table relational database instances from diverse public datasets, where these datasets are carefully curated to represent real-world scenarios incorporating varied table structures, relationships, and domains. **Second**, we introduce a novel sampling mechanism to create evaluation tasks with varying multi-table context lengths, ranging from 8K to 64K tokens - this mechanism enables us to assess the scalability of LLM's context length when processing multiple relation tables of different sizes, a critical requirement for real-world applications where data volumes can vary significantly. Adjustable context length is important for evaluating LLM's performance in its token limit. Meanwhile, we can mitigate contamination risk and make it easy to regenerate new evaluation splits if contamination is suspected. **Third**, to further reinforce the benchmark results' reliability, we incorporate symbolic extensions (Mirzadeh et al., 2024) into the evaluation framework, where flexible augmentation is integrated to evaluate the LLM's inherited reasoning ability over multi-table relational data instead of probabilistic retrieving or pattern matching. Both sampling and symbolic extension method makes our benchmark reliable enough and can be updated periodically. Comprehensively, we take a principled design to construct TQA-Bench, a multi-table QA benchmark, to evaluate LLMs' performance over complicated real-world relational QA tasks. Our concrete contribution can be summarized below:

- **A comprehensive multi-table QA benchmark.** We construct a new benchmark for multi-table QA that addresses the limitations of existing single-table benchmarks. Our benchmark incorporates varied relational data contexts by employing a sampling mechanism to generate evaluation tasks with context lengths ranging from 8K to 64K tokens. To ensure the reliability of evaluation results, we integrate symbolic extensions into the question templates, accessing the essential capabilities of LLMs beyond simple data retrieval or pattern matching.

- **A wide range of LLM evaluation results.** We systematically evaluate both open-source and closed-source LLMs on our benchmark, where the open-source models span a range of scales, from 2 billion to 671 billion parameters. We provide a comprehensive assessment of LLMs over real-world multi-table QA tasks.

- **Key observations and insights.** Our comprehensive evaluation yields the following key insights: (**i**) **Single- vs. multi-table performance**: LLMs consistently perform better in single-table settings, achieving up to $20\%$ higher accuracy compared to multi-table scenarios. This highlights the inadequacy of existing single-table benchmarks in capturing the complexity of real-world analytical tasks and underscores the need for dedicated multi-table QA evaluation. (**ii**) **Table serialization format**: The choice of serialization format significantly affects the model performance. Markdown outperforms CSV, JSON, and HTML across most LLMs and context lengths, providing a more LLM-friendly structure. (**iii**) **Model category and context sensitivity**: Instruction-tuned LLMs significantly outperform chat-oriented and domain-specific models, particularly under long-context settings. Reasoning LLMs (e.g., DEEPSEEK-R1) achieve state-of-the-art performance, while some distilled variants often fail to handle longer contexts. Overall, increasing context length leads to consistent performance degradation, especially for aggregation and complex analytical tasks. (**iv**) **Sampling and symbolic extension**: Our symbolic extension and database sampling strategies enhance benchmark diversity and robustness, which introduces a wider range of query patterns and difficulty levels, reduces variance in evaluation results, and enables consistent assessment across different database instances and question templates. (**v**) **Direct prompting vs. Text2SQL**: LLM-based Text2SQL methods demonstrate stable performance across context lengths and offer a complementary approach to direct prompting.

However, they struggle with complex analytical queries, such as correlation calculations, due to challenges in composing semantically correct SQL, and actually underperform the best LLMs by direct-prompting.

## 2 BENCHMARK CONSTRUCTION

We consider *multi-table QA* as the task of answering a single natural language question using tabular data from two or more distinct tables that are semantically related (e.g., through foreign-key relationships). The correct answer may require joining information across tables and possibly performing computations or analytics over the combined data.

The construction process of our multi-table QA benchmark is systematically divided into four key phases: data collection, relational data sampling, evaluation task definition, and question generation with symbolic extensions, where each phase can be summarized as:

- **Collect multi-table data** (§B.1). To ensure the diversity and representativeness of our benchmark, we collect a wide variety of large-scale relational databases. These databases serve as the foundation for generating multi-table QA tasks. We curate databases from three complementary sources: *World Bank*, *Data.gov*, and *BIRD*. World Bank contributes large, systematically related tables suitable for multi-table analytics; Data.gov offers heterogeneous public-sector datasets with diverse schemas; and BIRD provides relational environments originally developed for Text2SQL evaluation. We retain databases whose foreign-key graphs are valid so that sampled instances admit meaningful multi-table queries. Appendix §B.1 details the rationale of the sources, the filtering criteria, and the final database list.

- **Relational data sampling** (§B.2). We design a sampling methodology to create subsets of each table with varying serialized lengths. This approach ensures that the sampled data maintains the structural integrity and heterogeneity of the original datasets to evaluate LLM's performance under different context lengths. Raw databases can exceed LLM context budgets by orders of magnitude. We therefore create sampled database instances at target serialized lengths (e.g., 8k–64k tokens) via a two-stage procedure that **(i)** preserves foreign-key structure and **(ii)** approximates the token budget. Concretely, we topologically order tables on the foreign-key graph and sample rows parent-first; children are then restricted to referenced keys to maintain referential integrity. Next, we tune a sampling rate by binary search to hit the desired length; for each source database and each length, we generate multiple instances to reduce evaluation variance. Appendix §B.2 provides details and pseudocode, including the serializer used for token accounting.

- **Define evaluation task categories** (§B.3). We define three primary question categories, further divided into seven subcategories, inspired by those commonly found in traditional Table QA datasets. These categories are designed to capture a broad spectrum of question types, reflecting the diverse requirements of real-world multi-table QA tasks. Table 1 summarizes each subcategory with an informal relational-algebra (RA) sketch; complete formal definitions and examples are given in Appendix §B.3.

- **Generate question with symbolic extension** (§B.4). As illustrated in Figure 1, for each question subcategory, we develop structured question templates that are augmented with symbolic extensions to assess reasoning capabilities beyond simple retrieval. These templates are paired with Python-based answer generation, enabling the automated creation of benchmark questions

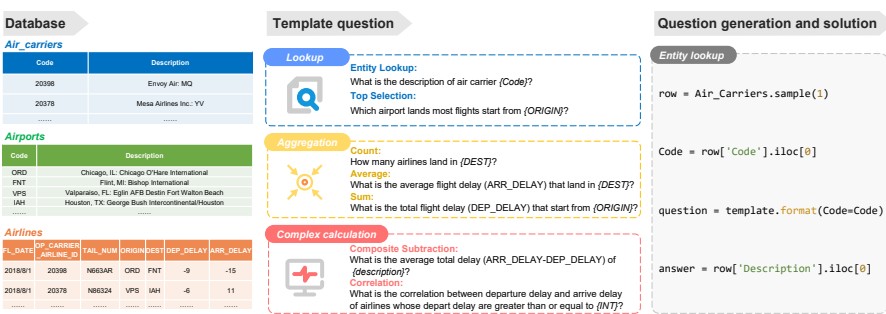

Figure 1: Symbolic extension formation in the "airline" database.

and ensuring scalability and reliability in task evaluation. Detailed generation method and more examples are listed in Appendix §B.4.

Table 1: Task category summary. RA uses selection $\sigma$, projection $\pi$, grouping $\gamma$, join $\bowtie$, and $\text{AGG} \in \{\text{COUNT}, \text{SUM}, \text{AVG}\}$. Given a condition $\Theta$, let $J$ denote the minimal join closure that contains all attributes required by $\Theta$ and the output.

| Primary | Subcategory | RA sketch |
|---|---|---|
| Lookup | Entity Lookup (EL) | $\pi_a \sigma_\Theta(J)$ |
| | Top Selection (TS) | $\pi_g \sigma_{c=\max \pi_c}(\gamma_{g;c:=\text{AGG}(e)}(\sigma_\Theta(J)))$ |
| Aggregation | Count (CNT) | $\text{COUNT}(\sigma_\Theta(J))$ |
| | Sum (SUM) | $\text{SUM}(\pi_a \sigma_\Theta(J))$ |
| | Average (AVG) | $\text{SUM}(\pi_a \sigma_\Theta(J))/\text{COUNT}(\sigma_\Theta(J))$ |
| Complex Calculation | Composite Comparison (CC) | $\text{AGG}(\pi_e \sigma_\Theta(J))$ |
| | Correlation (COR) | $\text{COR}(\pi_{\{a,b\}} \sigma_\Theta(J))$ |

This structured and systematic process enables the creation of a scalable, diverse, and effective benchmark for evaluating LLM performance on complex multi-table QA tasks.

## 3 EVALUATION SETUP

Building on our benchmark construction with database sampling and symbolic extension, we design experiments across multiple dimensions to systematically evaluate LLM performance.

**LLM Benchmark Scope**. To ensure a comprehensive evaluation of LLM's performance on our benchmark, we select 28 LLMs from various companies or research organizations. The selected models cover the most advanced proprietary LLMs available such as GPT (OpenAI, 2024c;b; 2025b; 2024d), as well as other widely-recognized open-source models such as QWEN2.5 (Team, 2024b). It is worth mentioning that we also choose a state-of-the-art model from DeepSeek that employs mixture-of-experts (MoE) architecture (DeepSeek-AI, 2024; Guo et al., 2025). Moreover, we include two domain-specific LLMs, TABLELLAMA (Zhang et al., 2024a) and TABLEGPT2 (Su et al., 2024), which are specifically fine-tuned for analyzing tabular data and accomplishing various table-based tasks. Meanwhile, the parameter scales of the models we choose range from 2B to 671B, which may provide insights into the relationship between model size and multi-table QA performance. Such diversity ensures the benchmark evaluates models of varying architectures, specializations, and computational complexities, providing valuable insights into the strengths and limitations of current LLMs in Table QA tasks. We enumerate the details of LLMs in the Appendix §C.

**Evaluation Design**. To better understand the performance of LLMs on multi-table QA, we propose the following research questions that capture different aspects of the challenge and motivate the design of our experiments. All subsequent experiments are constructed around these questions:

- (i) *What is the performance difference between single- and multi-table scenarios?*
- (ii) *How does the choice of table serialization format impact LLM performance in multi-table question answering tasks?*
- (iii) *How do different categories of LLMs vary in their ability to perform multi-table QA tasks under different context lengths?*
- (iv) *How does the symbolic extension, when combined with sampling, improve the diversity and difficulty of generated questions?*
- (v) *What is the performance variance between direct LLM prompt and LLM-based Text2SQL?*

## 4 EXPERIMENTS AND ANALYSIS

**Experiment Design**. To answer the five research questions in Section §3, we design a series of experiments (**Experiments 1–5**), each corresponding to one question. These are presented in Section 4.1 to Section 4.5, ensuring a clear one-to-one alignment between the evaluation questions and our experimental design. This step-by-step evaluation makes our benchmark rigorous, consistent, and provides meaningful insights into the performance of different LLMs in multi-table QA tasks. The experiment details, including dataset setup, evaluated LLMs, and model-specific observations, are provided in Appendix §D, and the prompts used in our evaluation are provided in Appendix §E. Moreover, the statistics of benchmark tasks are provided in Appendix §D.

## 4.1 COMPARE SINGLE- AND MULTI- TABLE SCENARIOS

**Experiment Setup.** The previous table QA benchmarks mainly focus on single-table settings. To examine how integrating multiple tables affects performance, we compare LLM accuracy in single-versus multi-table scenarios. In the single-table setting, only the necessary tables per query are merged into one table, while in the multi-table setting, the original relational structure is preserved.

**Comparison Results and Disscussion.** Table 2 shows that LLMs consistently achieve higher accuracy in the single-table setting, with up to 20% improvement. This confirms that single-table benchmarks cannot fully capture the complexity of real-world multi-table QA tasks and highlights the need for dedicated evaluation. Moreover, our analysis indicates that carefully designed merging strategies—for example, assigning semantic meaning to overlapping columns (e.g., renaming "`Description`" to "`ORIGIN_Description`" and "`DEST_Description`" when merging "`Airlines`" with "`Airports`") and restricting merges to relevant tables—can further boost performance and avoid excessive context length.

Table 2: Single-Table and multi-tables accuracy comparison.

| Model | GLM-4-9B-Chat | Qwen-2.5-7B-Instruct | Llama3.1-8B-Instruct | GPT-4o-mini | DS-R1-Qwen-7B |
|---|---|---|---|---|---|
| Single-Table | 32.43 | 55.36 | 49.21 | 54.50 | 34.00 |
| multi-tables | 22.14 | 35.29 | 31.79 | 48.43 | 28.79 |

> **Answer to Question (i)**: *Our findings show that LLMs generally outperformance in the single-table scenario, demonstrating that single-table benchmarks alone are insufficient for evaluating complex real-world Table QA applications comprehensively. At the same time, transforming multi-table inputs into single-table representations can be a potential avenue to improve model performance, provided that merging strategies preserve semantic integrity.*

## 4.2 SERIALIZATION FORMAT EVALUATION RESULTS

**Experiment Setup.** Given the importance of serialization in managing long-context, multi-table data, our first experiment compares four commonly used formats: Markdown, CSV, JSON and HTML. These formats are selected for their standardization. The goal is to identify the most effective format for subsequent experiments.

Before we start our experiment, we count the context length of different formats. Each format is serialized by using the pandas library. The results are in Table 3. Note that in terms of *serialization efficiencies* - the tokenized length after serialization into a given format, even for the same database, it varies significantly: CSV is the most compact format, whereas HTML takes nearly three times more tokens. In some cases, a 64K-scale database in HTML can exceed 128K tokens—beyond most LLM token limits. To ensure consistency, we tested HTML only up to a 32K scale. Table 4 summarizes the LLMs' performances across these formats.

Finally, we note that although all underlying databases in TQA-Bench are fully relational and equipped with explicit foreign-key graphs, the serialization formats induce different degrees of structural regularity in the model input. Markdown and CSV present tables as flat row–column grids, whereas JSON and HTML introduce more verbose, tree-like encodings. Across models and context lengths, JSON is consistently the weakest format and HTML often lags behind Markdown/CSV (Tables 3–4), suggesting that more semi-structured encodings of the same relational data can already make multi-table reasoning harder for current LLMs. At the same time, TQA-Bench does not yet include genuinely semi-structured table sources such as web tables or document-style JSON stores; extending our sampling and serialization pipeline to such heterogeneous data is an important direction for future table QA benchmarks.

Table 3: Context length of different formats and scales.

| Format | 8K | 16K | 32K | 64K |
|---|---|---|---|---|
| Markdown | $5.4 \times 10^3$ | $1.04 \times 10^4$ | $2.02 \times 10^4$ | $4.17 \times 10^4$ |
| CSV | $3.73 \times 10^3$ | $7.24 \times 10^3$ | $1.42 \times 10^4$ | $2.93 \times 10^4$ |
| JSON | $5.75 \times 10^3$ | $1.12 \times 10^4$ | $2.19 \times 10^4$ | $4.54 \times 10^4$ |
| HTML | $1.05 \times 10^4$ | $2.02 \times 10^4$ | $3.92 \times 10^4$ | $8.16 \times 10^4$ |

Table 4: Results of the serialization format comparison. Accuracies are computed at the question level. The "Average" column reports the macro-average across the seven subcategories—entity lookup (EL), top selection (TS), count (CNT), sum (SUM), average (AVG), composite comparison (CC), and correlation (COR).

| Model | Format | 8K | | | | | | | | 16K | | | | | | | |
|---|---|---|---|---|---|---|---|---|---|---|---|---|---|---|---|---|---|
| | | EL | TS | CNT | SUM | AVG | CC | COR | Average | EL | TS | CNT | SUM | AVG | CC | COR | Average |
| Qwen2.5-7B-Instruct | MD | 86.00 | 52.00 | 42.00 | 45.00 | 50.00 | 59.00 | 15.00 | 49.86 | 84.00 | 53.00 | 54.00 | 36.00 | 52.00 | 58.00 | 19.00 | 50.86 |
| | CSV | 75.00 | 36.00 | 25.00 | 38.00 | 38.00 | 61.00 | 11.00 | 40.57 | 71.00 | 41.00 | 30.00 | 27.00 | 24.00 | 51.00 | 14.00 | 36.86 |
| | JSON | 61.00 | 42.00 | 38.00 | 33.00 | 42.00 | 52.00 | 19.00 | 41.00 | 63.00 | 46.00 | 29.00 | 17.00 | 25.00 | 42.00 | 14.00 | 33.71 |
| | HTML | 82.00 | 43.00 | 40.00 | 33.00 | 41.00 | 66.00 | 21.00 | 46.57 | 78.00 | 55.00 | 29.00 | 24.00 | 26.00 | 52.00 | 31.00 | 42.14 |
| Qwen2.5-Coder-7B-Instruct | MD | 87.00 | 51.00 | 40.00 | 26.00 | 44.00 | 62.00 | 26.00 | 48.00 | 83.00 | 55.00 | 34.00 | 31.00 | 40.00 | 63.00 | 24.00 | 47.14 |
| | CSV | 88.00 | 50.00 | 43.00 | 32.00 | 42.00 | 62.00 | 16.00 | 47.57 | 80.00 | 43.00 | 30.00 | 28.00 | 38.00 | 59.00 | 23.00 | 43.00 |
| | JSON | 59.00 | 35.00 | 34.00 | 22.00 | 31.00 | 55.00 | 20.00 | 36.57 | 64.00 | 40.00 | 33.00 | 18.00 | 30.00 | 49.00 | 24.00 | 36.86 |
| | HTML | 81.00 | 45.00 | 38.00 | 27.00 | 39.00 | 61.00 | 23.00 | 44.86 | 81.00 | 47.00 | 31.00 | 24.00 | 39.00 | 54.00 | 26.00 | 43.14 |
| Llama3.1-8B-Instruct | MD | 86.00 | 62.00 | 32.00 | 30.00 | 44.00 | 60.00 | 12.00 | 46.57 | 84.00 | 58.00 | 29.00 | 20.00 | 34.00 | 56.00 | 20.00 | 43.00 |
| | CSV | 87.00 | 55.00 | 32.00 | 22.00 | 33.00 | 66.00 | 15.00 | 44.29 | 81.00 | 55.00 | 28.00 | 16.00 | 21.00 | 57.00 | 13.00 | 38.71 |
| | JSON | 65.00 | 37.00 | 26.00 | 19.00 | 24.00 | 49.00 | 20.00 | 34.29 | 70.00 | 53.00 | 19.00 | 12.00 | 17.00 | 45.00 | 10.00 | 32.29 |
| | HTML | 82.00 | 48.00 | 34.00 | 25.00 | 31.00 | 65.00 | 23.00 | 44.00 | 80.00 | 56.00 | 20.00 | 21.00 | 19.00 | 56.00 | 18.00 | 38.57 |
| | | **32K** | | | | | | | | **64K** | | | | | | | |
| Qwen2.5-7B-Instruct | MD | 67.00 | 51.00 | 32.00 | 28.00 | 33.00 | 45.00 | 35.00 | 41.57 | 44.00 | 49.00 | 26.00 | 22.00 | 26.00 | 32.00 | 35.00 | 33.43 |
| | CSV | 66.00 | 41.00 | 15.00 | 10.00 | 15.00 | 52.00 | 37.00 | 33.71 | 58.00 | 38.00 | 23.00 | 18.00 | 12.00 | 42.00 | 37.00 | 32.57 |
| | JSON | 44.00 | 39.00 | 21.00 | 14.00 | 19.00 | 27.00 | 26.00 | 27.14 | 41.77 | 62.03 | 26.58 | 13.92 | 15.19 | 28.21 | 21.79 | 29.95 |
| | HTML | 62.00 | 41.00 | 28.00 | 20.00 | 24.00 | 39.00 | 38.00 | 36.00 | OOC | OOC | OOC | OOC | OOC | OOC | OOC | OOC |
| Qwen2.5-Coder-7B-Instruct | MD | 75.00 | 51.00 | 33.00 | 24.00 | 38.00 | 51.00 | 42.00 | 44.86 | 63.00 | 50.00 | 20.00 | 19.00 | 31.00 | 40.00 | 35.00 | 36.86 |
| | CSV | 76.00 | 40.00 | 17.00 | 20.00 | 23.00 | 38.00 | 27.00 | 34.43 | 60.00 | 43.00 | 26.00 | 19.00 | 32.00 | 38.00 | 34.00 | 36.00 |
| | JSON | 49.00 | 43.00 | 30.00 | 20.00 | 20.00 | 39.00 | 33.00 | 33.43 | 45.00 | 43.00 | 23.00 | 12.00 | 25.00 | 20.00 | 27.00 | 27.86 |
| | HTML | 66.00 | 46.00 | 22.00 | 24.00 | 39.00 | 47.00 | 34.00 | 39.71 | OOC | OOC | OOC | OOC | OOC | OOC | OOC | OOC |
| Llama3.1-8B-Instruct | MD | 78.00 | 49.00 | 20.00 | 12.00 | 17.00 | 50.00 | 22.00 | 35.43 | 74.00 | 54.00 | 21.00 | 10.00 | 18.00 | 45.00 | 29.00 | 35.86 |
| | CSV | 73.00 | 50.00 | 19.00 | 15.00 | 9.00 | 44.00 | 21.00 | 33.00 | 75.00 | 55.00 | 21.00 | 17.00 | 11.00 | 41.00 | 30.00 | 35.71 |
| | JSON | 51.00 | 42.00 | 14.00 | 12.00 | 9.00 | 37.00 | 14.00 | 25.57 | 57.47 | 45.98 | 16.28 | 13.95 | 10.47 | 29.07 | 19.77 | 27.65 |
| | HTML | 75.00 | 48.00 | 15.00 | 13.00 | 18.00 | 54.00 | 32.00 | 36.43 | OOC | OOC | OOC | OOC | OOC | OOC | OOC | OOC |

**Overall Results.** The benchmark results indicate that *Markdown consistently leads better performances other formats across a majority of LLMs*. While CSV and HTML show no clear advantage over each other, JSON is the weakest, yielding the lowest accuracy in almost all scales and models.

**Detailed Discussion.** We further enumerate several interesting observations. **First**, *Markdown's advantage holds across most subcategories and context lengths, with only minor deviations*. **Second**, *our analysis revealed that coder LLMs outperformed their original counterparts with certain formats*. **Overall**, based on these findings, we adopt Markdown as the standard format in subsequent experiments to ensure consistency and optimal performance.

> **Answer to Question (ii)**: *Our findings suggest that Markdown is the superior format for table serialization for LLMs in multi-table QA tasks compared with CSV, JSON, and HTML.*

### 4.3 COMPREHENSIVE LLM EVALUATION RESULTS

**Experiment Setup.** Having selected Markdown as the serialization format, we undertake a comprehensive evaluation of LLMs from various providers and of different scales. This experiment allows us to observe performance trends and compare capabilities under uniform conditions across different LLMs. Table 5 and Figure 3 summarize the concrete evaluation results and overall performance comparison of LLMs. We enumerate our interesting observations based on LLM categories:

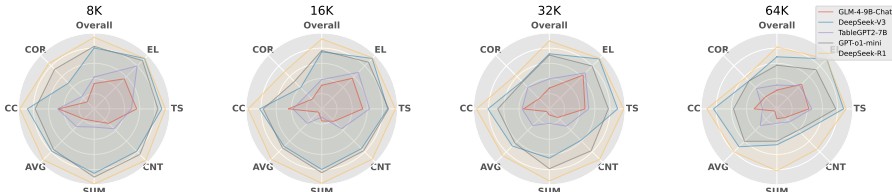

Figure 2: The accuracy distribution of question subcategories in different context lengths.

**Chat LLM Performance.** *Chat-oriented models generally underperform, with most achieving less than 25% accuracy due to weak instruction adherence.* They often output invalid responses such as "I don't know", "None of the above", multiple answers, or verbose explanations that deviate from the required format which has been explicitly clarified in the instruction. We speculate this behavior likely stems from their design, which prioritizes conversational fluency over strict adherence to task-specific instructions. Interestingly, we also find that *the overall accuracy of chat models tends to decline as their scale increases*.

**Instruct LLM Performance.** We find that *instruct LLMs demonstrated superior adherence to instructions and generally perform better, with most exceeding 25%*. As expected, larger instruct models generally performed better, showing a clear trend of improved accuracy with scale. Notably,

Table 5: Complete benchmark results. NFI indicates "not following instructions", and OOC indicates "out of context".

| | 8K | | | | | | | | 16K | | | | | | | |
|---|---|---|---|---|---|---|---|---|---|---|---|---|---|---|---|---|
| Model | EL | TS | CNT | SUM | AVG | CC | COR | Average | EL | TS | CNT | SUM | AVG | CC | COR | Average |
| **Chat Models** | | | | | | | | | | | | | | | | |
| GLM-4-9B-Chat | 56.00 | 56.00 | 27.00 | 16.00 | 19.00 | 47.00 | 13.00 | 33.43 | 57.00 | 54.00 | 23.00 | 16.00 | 6.00 | 44.00 | 18.00 | 31.14 |
| Baichuan2-7B-Chat | 35.00 | 34.00 | 22.00 | 19.00 | 20.00 | 17.00 | 22.00 | 24.14 | 26.00 | 45.00 | 14.00 | 11.00 | 14.00 | 17.00 | 16.00 | 20.43 |
| Baichuan2-13B-Chat | 2.00 | 4.00 | 8.00 | 2.00 | 4.00 | 3.00 | 7.00 | 4.29 | 5.00 | 3.00 | 1.00 | 4.00 | 3.00 | 4.00 | 5.00 | 3.86 |
| Vicuna-7B-V1.5-16K | 29.00 | 22.00 | 40.00 | 20.00 | 23.00 | 30.00 | 27.00 | 27.29 | 11.00 | 23.00 | 19.00 | 11.00 | 13.00 | 13.00 | 8.00 | 14.00 |
| Vicuna-13B-V1.5-16K | 31.00 | 29.00 | 12.00 | 2.00 | 8.00 | 10.00 | 12.00 | 14.86 | 19.00 | 26.00 | 19.00 | 3.00 | 7.00 | 9.00 | 4.00 | 12.43 |
| **Instruct Models** | | | | | | | | | | | | | | | | |
| Mistral-7B-Instruct | 44.00 | 41.00 | 16.00 | 8.00 | 13.00 | 25.00 | 12.00 | 22.71 | 46.00 | 48.00 | 14.00 | 5.00 | 10.00 | 18.00 | 4.00 | 20.71 |
| Mistral-Nemo-Instruct | 83.00 | 59.00 | 38.00 | 32.00 | 40.00 | 57.00 | 33.00 | 48.86 | 42.00 | 43.00 | 21.00 | 13.00 | 17.00 | 34.00 | 18.00 | 26.86 |
| Llama3.1-8B-Instruct | 86.00 | 62.00 | 32.00 | 30.00 | 44.00 | 60.00 | 12.00 | 46.57 | 84.00 | 58.00 | 29.00 | 20.00 | 34.00 | 56.00 | 20.00 | 43.00 |
| Llama3.1-70B-Instruct | 93.00 | 82.00 | 57.00 | 51.00 | 54.00 | 83.00 | 20.00 | 62.86 | 94.00 | 81.00 | 39.00 | 39.00 | 46.00 | 80.00 | 20.00 | 57.00 |
| Qwen2.5-3B-Instruct | 62.00 | 31.00 | 23.00 | 24.00 | 23.00 | 46.00 | 8.00 | 31.00 | 59.00 | 36.00 | 25.00 | 15.00 | 25.00 | 36.00 | 12.00 | 29.71 |
| Qwen2.5-7B-Instruct | 86.00 | 52.00 | 42.00 | 45.00 | 50.00 | 59.00 | 15.00 | 49.86 | 84.00 | 53.00 | 54.00 | 36.00 | 52.00 | 58.00 | 19.00 | 50.86 |
| Qwen2.5-Coder-7B-Instruct | 87.00 | 51.00 | 40.00 | 26.00 | 44.00 | 62.00 | 26.00 | 48.00 | 83.00 | 55.00 | 34.00 | 31.00 | 40.00 | 63.00 | 24.00 | 47.14 |
| Qwen2.5-14B-Instruct | 89.00 | 72.00 | 60.00 | 47.00 | 59.00 | 80.00 | 9.00 | 59.43 | 86.00 | 75.00 | 48.00 | 35.00 | 43.00 | 72.00 | 13.00 | 53.14 |
| Qwen2.5-72B-Instruct | 91.00 | 72.00 | 55.00 | 59.00 | 51.00 | 78.00 | 3.00 | 58.43 | 86.00 | 61.00 | 34.00 | 34.00 | 38.00 | 73.00 | 1.00 | 46.71 |
| Gemma2-2B-It | 49.00 | 33.00 | 28.00 | 25.00 | 13.00 | 18.00 | 31.00 | 28.14 | OOC | OOC | OOC | OOC | OOC | OOC | OOC | OOC |
| Gemma2-9B-It | 76.00 | 41.00 | 42.42 | 28.72 | 27.96 | 46.32 | 2.11 | 38.17 | OOC | OOC | OOC | OOC | OOC | OOC | OOC | OOC |
| Gemma2-27B-It | 83.00 | 44.00 | 29.00 | 31.00 | 33.00 | 65.00 | 15.00 | 42.86 | OOC | OOC | OOC | OOC | OOC | OOC | OOC | OOC |
| DeepSeek-V3 | 94.00 | 89.00 | 79.00 | 84.85 | 79.00 | 88.00 | 48.00 | 80.26 | 94.00 | 86.87 | 74.00 | 79.80 | 70.00 | 81.82 | 39.39 | 75.14 |
| **Table Specific Models** | | | | | | | | | | | | | | | | |
| TableGPT2-7B | 80.00 | 49.00 | 37.00 | 24.00 | 33.00 | 48.00 | 23.00 | 42.00 | 68.00 | 63.00 | 37.00 | 11.00 | 27.00 | 38.00 | 27.00 | 38.71 |
| TableLlama | NFI | NFI | NFI | NFI | NFI | NFI | NFI | NFI | OOC | OOC | OOC | OOC | OOC | OOC | OOC | OOC |
| **Close-Source Models** | | | | | | | | | | | | | | | | |
| GPT-4o-mini | 82.00 | 66.00 | 55.00 | 51.00 | 60.00 | 72.00 | 40.00 | 60.86 | 82.00 | 71.00 | 49.00 | 47.00 | 57.00 | 68.00 | 24.00 | 56.86 |
| GPT-4o | 92.00 | 88.00 | 80.00 | 76.00 | 76.00 | 90.00 | 48.48 | 78.68 | 91.00 | 82.00 | 75.00 | 63.00 | 68.00 | 85.00 | 42.00 | 72.29 |
| GPT-o1-mini | 90.00 | 84.00 | 85.00 | 89.90 | 75.76 | 77.78 | 73.74 | 82.33 | 87.00 | 88.00 | 78.00 | 84.00 | 73.00 | 77.00 | 51.00 | 76.86 |
| GPT-o3-mini | 89.00 | 86.00 | 91.92 | 93.00 | 85.00 | 91.00 | 86.00 | 88.84 | 92.00 | 91.00 | 94.00 | 94.00 | 84.00 | 93.00 | 84.00 | 90.29 |
| **Reasoning Models** | | | | | | | | | | | | | | | | |
| DeepSeek-R1-Distill-Qwen-7B | 39.00 | 40.00 | 39.00 | 15.00 | 20.00 | 21.00 | 21.00 | 27.86 | 29.00 | 31.00 | 19.00 | 14.00 | 16.00 | 28.00 | 25.00 | 23.14 |
| DeepSeek-R1-Distill-Qwen-14B | 89.00 | 59.00 | 52.00 | 48.00 | 44.00 | 71.00 | 9.00 | 53.14 | 83.00 | 59.00 | 37.00 | 28.00 | 24.00 | 58.00 | 13.00 | 43.14 |
| DeepSeek-R1 | 94.00 | 93.94 | 95.96 | 98.99 | 94.95 | 98.00 | 84.69 | 94.38 | 94.95 | 97.00 | 93.94 | 98.00 | 94.00 | 97.00 | 69.39 | 92.10 |
| QwQ-32B-Preview | 1.00 | 4.00 | 2.00 | 2.00 | 1.00 | 2.00 | 2.00 | 2.00 | 1.00 | 3.00 | 1.00 | 0.00 | 1.00 | 4.00 | 3.00 | 2.00 |

| | 32K | | | | | | | | 64K | | | | | | | |
|---|---|---|---|---|---|---|---|---|---|---|---|---|---|---|---|---|
| Model | EL | TS | CNT | SUM | AVG | CC | COR | Average | EL | TS | CNT | SUM | AVG | CC | COR | Average |
| **Chat Models** | | | | | | | | | | | | | | | | |
| GLM-4-9B-Chat | 63.00 | 47.00 | 16.00 | 8.00 | 5.00 | 34.00 | 19.00 | 27.43 | 46.00 | 42.00 | 16.00 | 13.00 | 4.00 | 30.00 | 22.00 | 24.71 |
| Baichuan2-7B-Chat | 27.00 | 33.00 | 12.00 | 13.00 | 14.00 | 13.00 | 12.00 | 17.71 | 28.00 | 41.00 | 21.00 | 16.00 | 22.00 | 14.00 | 17.00 | 22.71 |
| Baichuan2-13B-Chat | 4.00 | 4.00 | 4.00 | 4.00 | 1.00 | 3.00 | 1.00 | 3.00 | 5.00 | 5.00 | 5.00 | 5.00 | 1.00 | 3.00 | 1.00 | 3.57 |
| Vicuna-7B-V1.5-16K | OOC | OOC | OOC | OOC | OOC | OOC | OOC | OOC | OOC | OOC | OOC | OOC | OOC | OOC | OOC | OOC |
| Vicuna-13B-V1.5-16K | OOC | OOC | OOC | OOC | OOC | OOC | OOC | OOC | OOC | OOC | OOC | OOC | OOC | OOC | OOC | OOC |
| **Instruct Models** | | | | | | | | | | | | | | | | |
| Mistral-7B-Instruct | 43.00 | 35.00 | 10.00 | 7.00 | 5.00 | 9.00 | 9.00 | 16.86 | OOC | OOC | OOC | OOC | OOC | OOC | OOC | OOC |
| Mistral-Nemo-Instruct | 19.00 | 22.00 | 6.00 | 11.00 | 11.00 | 15.00 | 14.00 | 14.00 | 6.00 | 12.00 | 6.00 | 1.00 | 7.00 | 5.00 | 6.00 | 6.14 |
| Llama3.1-8B-Instruct | 78.00 | 49.00 | 20.00 | 12.00 | 17.00 | 50.00 | 22.00 | 35.43 | 74.00 | 54.00 | 21.00 | 10.00 | 18.00 | 45.00 | 29.00 | 35.86 |
| Llama3.1-70B-Instruct | 93.00 | 73.00 | 29.00 | 24.00 | 33.00 | 76.00 | 27.00 | 50.71 | 88.00 | 83.00 | 26.00 | 25.00 | 27.00 | 57.00 | 29.00 | 47.86 |
| Qwen2.5-3B-Instruct | 46.00 | 28.00 | 19.00 | 11.00 | 21.00 | 22.00 | 15.00 | 23.14 | 39.00 | 33.00 | 11.00 | 10.00 | 19.00 | 24.00 | 19.00 | 22.14 |
| Qwen2.5-7B-Instruct | 67.00 | 51.00 | 32.00 | 28.00 | 33.00 | 45.00 | 35.00 | 41.57 | 44.00 | 49.00 | 26.00 | 22.00 | 26.00 | 32.00 | 35.00 | 33.43 |
| Qwen2.5-Coder-7B-Instruct | 75.00 | 51.00 | 33.00 | 24.00 | 38.00 | 51.00 | 42.00 | 44.86 | 63.00 | 50.00 | 20.00 | 19.00 | 31.00 | 40.00 | 35.00 | 36.86 |
| Qwen2.5-14B-Instruct | 83.00 | 67.00 | 27.00 | 24.00 | 21.00 | 59.00 | 8.00 | 41.29 | 50.00 | 56.00 | 25.00 | 15.00 | 14.00 | 28.00 | 23.00 | 30.14 |
| Qwen2.5-72B-Instruct | 85.00 | 57.00 | 16.00 | 26.00 | 18.00 | 66.00 | 4.00 | 38.86 | 69.00 | 51.00 | 14.00 | 13.00 | 13.13 | 54.00 | 4.00 | 31.19 |
| Gemma2-2B-It | OOC | OOC | OOC | OOC | OOC | OOC | OOC | OOC | OOC | OOC | OOC | OOC | OOC | OOC | OOC | OOC |
| Gemma2-9B-It | OOC | OOC | OOC | OOC | OOC | OOC | OOC | OOC | OOC | OOC | OOC | OOC | OOC | OOC | OOC | OOC |
| Gemma2-27B-It | OOC | OOC | OOC | OOC | OOC | OOC | OOC | OOC | OOC | OOC | OOC | OOC | OOC | OOC | OOC | OOC |
| DeepSeek-V3 | 93.00 | 90.00 | 58.76 | 64.95 | 69.07 | 80.81 | 51.00 | 72.61 | 93.00 | 87.88 | 44.90 | 47.31 | 70.53 | 83.51 | 52.00 | 68.62 |
| **Table Specific Models** | | | | | | | | | | | | | | | | |
| TableGPT2-7B | 67.00 | 52.00 | 32.00 | 19.00 | 31.00 | 36.00 | 40.00 | 39.57 | 43.00 | 46.00 | 24.00 | 19.00 | 31.00 | 19.00 | 38.00 | 31.43 |
| TableLlama | OOC | OOC | OOC | OOC | OOC | OOC | OOC | OOC | OOC | OOC | OOC | OOC | OOC | OOC | OOC | OOC |
| **Close-Source Models** | | | | | | | | | | | | | | | | |
| GPT-4o-mini | 82.00 | 68.00 | 44.00 | 38.00 | 59.00 | 63.00 | 38.00 | 56.00 | 74.00 | 73.00 | 31.00 | 34.00 | 47.00 | 58.00 | 36.73 | 50.57 |
| GPT-4o | 88.00 | 93.00 | 56.00 | 54.00 | 60.00 | 81.00 | 50.00 | 68.86 | 90.62 | 85.42 | 46.74 | 32.63 | 54.95 | 83.33 | 47.78 | 63.41 |
| GPT-o1-mini | 79.80 | 77.78 | 78.00 | 78.79 | 64.00 | 68.00 | 50.00 | 70.88 | 73.00 | 77.55 | 39.80 | 42.55 | 60.42 | 57.14 | 52.00 | 57.60 |
| GPT-o3-mini | 89.00 | 87.00 | 91.00 | 96.00 | 89.00 | 87.00 | 63.00 | 86.00 | 80.00 | 82.83 | 78.00 | 78.00 | 75.00 | 74.00 | 53.54 | 74.50 |
| **Reasoning Models** | | | | | | | | | | | | | | | | |
| DeepSeek-R1-Distill-Qwen-7B | 25.00 | 27.00 | 9.00 | 8.00 | 11.00 | 18.00 | 23.00 | 17.29 | NFI | NFI | NFI | NFI | NFI | NFI | NFI | NFI |
| DeepSeek-R1-Distill-Qwen-14B | 78.00 | 49.00 | 18.00 | 12.00 | 8.00 | 42.00 | 15.00 | 31.71 | 57.00 | 37.00 | 20.00 | 14.00 | 15.00 | 24.00 | 31.00 | 28.29 |
| DeepSeek-R1 | 94.00 | 98.00 | 94.00 | 94.95 | 87.00 | 96.00 | 63.00 | 89.56 | 92.63 | 91.11 | 75.00 | 81.61 | 80.22 | 92.31 | 58.33 | 81.50 |
| QwQ-32B-Preview | 3.00 | 2.00 | 1.00 | 0.00 | 2.00 | 2.00 | 9.00 | 2.71 | OOC | OOC | OOC | OOC | OOC | OOC | OOC | OOC |

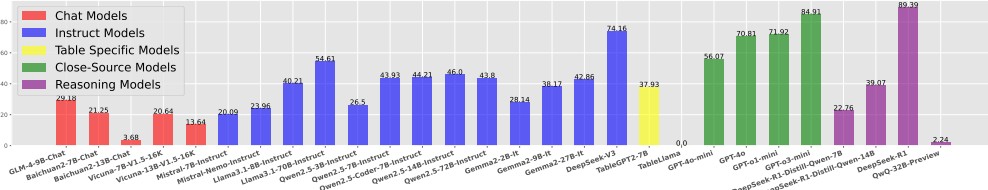

Figure 3: The overall accuracy of all models.

coding LLMs within this category performed slightly better than their standard instruction-following counterparts. However, the performance gap was relatively marginal, suggesting that this observation may not generalize across all tasks or datasets.

**Domain-Specific Tabular LLMs.** We find that *LLMs fine-tuned for relational tasks, such as* TABLELLAMA *and* TABLEGPT2, *underperform compared to expectations.* TABLELLAMA fails to follow the required answer format entirely, while TABLEGPT2 adheres more consistently but achieves only average accuracy.

**Reasoning LLM Performance.** Our experimental results reveal *notable disparities in the performance of reasoning models on our benchmark*. Notably, DEEPSEEK-R1 demonstrates state-of-the-art performance across all evaluated scales. While reasoning models are often assumed to excel

at logical and analytical tasks, our findings suggest that this expectation does not hold for distilled models. Specifically, distilled versions fail to surpass their original counterparts, like the distilled QWEN models, which perform worse than the original ones.

**Impact of Context Length.** We analyze how table context length affects the performance of LLMs, focusing on instruct LLMs. Our experiments show that *performance generally decreases as context length increases*. While some LLMs show slight improvements with longer contexts, these gains are marginal. Even in simpler subcategories, performance declines as table context grows, confirming that handling long tabular contexts remains challenging for all but the simplest questions. Detailed task-specific observations are provided in Appendix §D.3, and Figure 2 illustrates this phenomenon.

> **Answer to Question (iii)**: *Our comprehensive evaluation highlights interesting observations: instruct-tuned LLMs exhibit significantly better task performance compared to chat-oriented LLMs. Specialized tabular LLMs demonstrate limited flexibility, underperforming relative to expectations. Reasoning LLMs can exhibit optimal performance (i.e., DEEPSEEK-R1), whereas distillation may negatively impact accuracy and long-context handling. Moreover, longer context lengths consistently challenge LLMs, with substantial performance drops in aggregation and complex calculation tasks.*

## 4.4 SAMPLING AND SYMBOLIC EXTENSION

**Experiment Setup.** This experiment evaluates how sampling and symbolic extensions increase the diversity and difficulty of the benchmark, thereby improving its ability to assess LLM analytic capabilities. Detailed procedures for question generation and batch construction are provided in Appendix §D.4. We visualize results using heatmaps (batch-level accuracy of question instances) and histograms (accuracy distributions across batches). We also compare average accuracy across three test sets: all airline batches, five airline batches (5 in airline), and five batches sampled across all databases (5 in all), as shown in Figure 4.

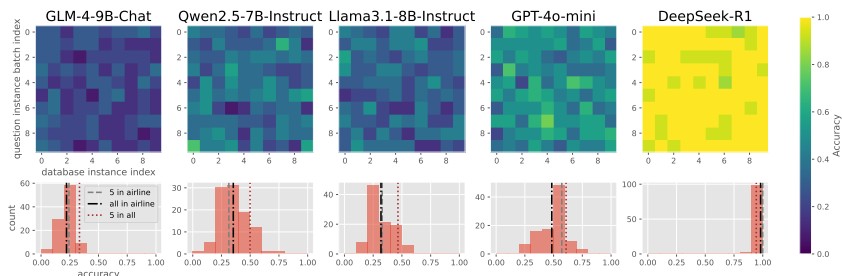

Figure 4: Accuracy distribution of 8K airline database question instances across four models.

**Results and Detailed Analysis.** The heatmaps reveal that question difficulty varies across batches and models, indicating that different LLMs have distinct preferences when handling certain question instances. Incorporating sampling and symbolic extensions thus enhances the stability of the benchmark, ensuring more reliable assessment of model performance. The histograms further show that too few questions yield unstable results, underscoring the value of sampling and symbolic extensions for a more balanced evaluation. Finally, results are consistent between broad and sensitive tests in the "airline" database, suggesting that using five batches, as in earlier studies, is often sufficient to approximate performance across the full dataset. A broader comparison across all databases versus the airline database also reveals that question difficulty differs across datasets.

> **Answer to Question (iv)**: *The sampling mechanism combined with sampling and symbolic extension will generate reliable benchmark results, where when the number of sampled database instances increases, it will lead to different sampling and symbolic extension executions. The alignment of the accuracy distribution illustrates the stable benchmark results when equipped with both our sampling mechanism and symbolic extensions.*

### 4.5 DIRECT LLM PROMPT VS. TEXT2SQL

**Experiment Setup.** Given the prevalence of relational databases in practice, Text2SQL has emerged as a widely used paradigm. We compare direct LLM prompting with LLM-based Text2SQL to identify which approach better handles multi-table QA.

Table 6: Text2SQL Performance in the TQA-Bench.

| Model | 8K | | | | | | | | 16K | | | | | | | |
|---|---|---|---|---|---|---|---|---|---|---|---|---|---|---|---|---|
| | EL | TS | CNT | SUM | AVG | CC | COR | Average | EL | TS | CNT | SUM | AVG | CC | COR | Average |
| GLM-4-9B-Chat | 54.00 | 27.00 | 46.00 | 39.00 | 25.00 | 20.00 | 0.00 | 30.14 | 49.00 | 21.00 | 47.00 | 28.00 | 25.00 | 21.00 | 0.00 | 27.29 |
| Qwen2.5-Coder-7B-Instruct | 31.00 | 20.00 | 32.00 | 33.00 | 21.00 | 19.00 | 0.00 | 22.29 | 40.00 | 16.00 | 34.00 | 31.00 | 26.00 | 13.00 | 0.00 | 22.86 |
| Llama3.1-8B-Instruct | 39.00 | 11.00 | 33.00 | 24.00 | 27.00 | 13.00 | 0.00 | 21.00 | 33.00 | 12.00 | 28.00 | 29.00 | 18.00 | 6.00 | 0.00 | 18.00 |
| Arctic-Text2SQL-R1-7B | 74.00 | 50.00 | 78.00 | 57.00 | 62.00 | 43.00 | 3.00 | 52.43 | 75.00 | 43.00 | 72.00 | 59.00 | 58.00 | 37.00 | 2.00 | 49.43 |
| GPT-4o-mini | 74.00 | 55.00 | 77.00 | 55.00 | 56.00 | 42.00 | 0.00 | 51.29 | 78.00 | 45.00 | 72.00 | 56.00 | 54.00 | 34.00 | 0.00 | 48.43 |
| DeepSeek-R1 | 85.00 | 73.00 | 80.00 | 73.00 | 64.00 | 52.00 | 14.00 | 63.00 | 87.00 | 69.00 | 78.00 | 70.00 | 58.00 | 54.00 | 20.00 | 62.29 |
| | 32K | | | | | | | | 64K | | | | | | | |
| GLM-4-9B-Chat | 53.00 | 30.00 | 48.00 | 31.00 | 24.00 | 21.00 | 0.00 | 29.57 | 50.00 | 26.00 | 44.00 | 29.00 | 22.00 | 36.00 | 0.00 | 29.57 |
| Qwen2.5-Coder-7B-Instruct | 27.00 | 24.00 | 38.00 | 19.00 | 22.00 | 11.00 | 1.00 | 20.29 | 39.00 | 22.00 | 30.00 | 21.00 | 18.00 | 18.00 | 0.00 | 21.14 |
| Llama3.1-8B-Instruct | 42.00 | 12.00 | 33.00 | 24.00 | 19.00 | 9.00 | 0.00 | 19.86 | 29.00 | 11.00 | 28.00 | 21.00 | 12.00 | 11.00 | 0.00 | 16.00 |
| Arctic-Text2SQL-R1-7B | 75.00 | 48.00 | 74.00 | 48.00 | 52.00 | 43.00 | 1.00 | 48.71 | 74.00 | 43.00 | 61.00 | 48.00 | 50.00 | 41.00 | 3.00 | 45.71 |
| GPT-4o-mini | 73.00 | 54.00 | 74.00 | 49.00 | 49.00 | 40.00 | 0.00 | 48.43 | 74.00 | 49.00 | 65.00 | 47.00 | 47.00 | 31.00 | 0.00 | 44.71 |
| DeepSeek-R1 | 84.00 | 74.00 | 73.00 | 63.00 | 50.00 | 50.00 | 11.00 | 57.86 | 84.00 | 65.00 | 66.00 | 63.00 | 46.00 | 54.00 | 8.00 | 55.14 |

**Results and Detailed Analysis.** The results of LLM-based Text2SQL are listed in Table 6. Comparison with the comprehensive LLM prompt results (Table 5) reveals a few interesting observations: **First**, we find significant distinctions between these two approaches concerning context-length sensitivity: *While the performance of direct prompt tends to deteriorate notably with increasing table lengths due to the escalating complexity of inputs, the performance of LLM-based Text2SQL remains relatively stable across context lengths ranging from 8K to 64K tokens.* Specifically, top-performing models like DEEPSEEK-R1 exhibited only slight performance declines as context increased, demonstrating the robustness of schema-only prompting against context length variation. **Second**, despite their stability in handling varied context lengths, *LLM-based Text2SQL methods still faced notable challenges with complex analytical queries, particularly correlation tasks, worse than direct LLM prompt methods.* This mirrors the limitations identified in end-to-end TableQA experiments, highlighting that advanced analysis could remain challenging for both approaches.

> **Answer to Question (v)**: *LLM-based Text2SQL methods offer stable performance across varying context lengths, complementing direct prompting approaches. However, they struggle with complex analytical tasks, often generating incorrect SQL and falling short of the peak performance achieved by top LLMs using direct prompting.*

## 5 RELATED WORK

**Table QA.** Question answering (QA) over relational databases has long been central to natural language processing and data management (Jin et al., 2022). Given a user query, table QA seeks accurate answers via table understanding and reasoning (Pal et al., 2023; Zhang et al., 2024b; Zhu et al., 2024). Methods are commonly grouped into two classes: (i) *Text2SQL*, which translates natural language into executable SQL (Liu et al., 2021; Fu et al., 2023; Gu et al., 2023; Li et al., 2024a; Fan et al., 2024; Zhang et al., 2024b; Katsogiannis-Meimarakis & Koutrika, 2023); and (ii) *end-to-end* models that process the question with a serialized table to directly produce an answer (Nan et al., 2022; Zhao et al., 2022; Cheng et al., 2021; Chen et al., 2020a;c). Representative E2E systems include TABLE-BERT (Chen et al., 2020b), which converts tables into coherent text for downstream processing; TAPAS (Herzig et al., 2020), which encodes tables within BERT; and PASTA (Gu et al., 2022), which pre-trains on cloze-style sentence–table tasks using WikiTables. Multi-table QA has been explored by MULTITABQA (Pal et al., 2023), while AutoTQA (Zhu et al., 2024) uses multi-agent LLMs for conversational solving. Our benchmark focuses on comprehensively evaluating techniques in the second class with reliable, comparable results.

**Assessment of LLM over data management tasks.** LLMs enable new AI applications (Bommasani et al., 2021) and are reshaping data management (Biswal et al., 2024; Chen et al., 2024; Patel et al., 2024; Wornow et al., 2024), including data integration (Huo et al., 2024; Döhmen et al., 2024), tuning (Giannakouris & Trummer, 2024), query optimization (Liu et al., 2024), table summarization (Liu et al., 2022), and formatting (Singh et al., 2023). Domain-specific table LLMs: TABLELLAMA (Zhang et al., 2023) (fine-tuned on TableInstruct for multiple in-domain tasks) and TABLEGPT (Zha et al., 2023; Su et al., 2024) (unifying tables, language, and commands for QA, manipulation, visualization, and reporting). Benchmarks evaluate Text2SQL (Yu et al., 2018; Lei et al., 2024; Gao et al., 2024), relational structure understanding (Sui et al., 2024), and table QA (Chen et al., 2020b; Lei et al., 2023; Wu et al., 2025b). We introduce a multi-table QA

benchmark to assess LLM reasoning robustly with variable relational contexts and symbolic extension.

## 6 CONCLUSION

In this paper, we introduce TQA-Bench, a new multi-table QA benchmark specifically designed to rigorously evaluate the capabilities of LLMs in processing complex, relational data across multiple tables. Our benchmark applies diverse relational database instances drawn from real-world public datasets, a flexible sampling mechanism that allows for the creation of tasks with varying context lengths from 8K to 64K tokens, and the integration of symbolic extensions to test higher-order reasoning capabilities. Through systematic evaluations involving both open-source and closed-source LLMs, with scales ranging from 2 billion to 671 billion parameters, our findings highlight the variable performance of these models under complex multi-table QA scenarios. We expect that TQA-Bench can serve as a pivotal step toward realizing the full potential of LLMs in the analysis of complex tabular data.

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

## A CURRENT TABLE QA BENCHMARKS

We collect mainstream table QA benchmarks and summarize them in Table 7. Most of these benchmarks are built on Wikipedia tables, reflecting that Wikipedia is both a widely used benchmark source and a common component of LLM pre-training corpora. For TQA-Bench, we deliberately exclude Wikipedia tables and instead construct our benchmark from non-Wikipedia sources (World-Bank, Data.gov, and BIRD), which reduces reliance on the most heavily reused Wikipedia tables and provides large relational schemas that better match our multi-table evaluation setting. However, this choice alone cannot fully prevent pre-training contamination, since these public datasets may also appear in model training data. In this work we do not perform a full pre-training-corpus contamination analysis—such an analysis is infeasible for proprietary LLMs—and our design should therefore be viewed as mitigating contamination risk and enabling easy regeneration of fresh evaluation splits, rather than eliminating contamination entirely. To that end, we combine relational sampling with symbolic extensions so that (i) questions and answers are generated by sampling relational subgraphs and recomputing derived quantities, making exact question–answer pairs unlikely to coincide with memorized facts, and (ii) the entire pipeline can be re-run with new random seeds to regenerate new database instances and evaluation splits if contamination is suspected.

Below we additionally discuss two very recent multi-table QA benchmarks, MMQA and MTab-VQA, which are closely related to TQA-Bench but are not included in the table.

**MMQA (Wu et al., 2025a).** MMQA evaluates LLMs on multi-table, multi-hop questions built on top of the Spider databases. Conceptually, it is close to our work in that it targets complex questions over multiple relational tables. However, MMQA remains within the Spider schema, whereas TQA-Bench constructs diverse analytical databases from WorldBank, Data.gov, and BIRD, which differ in domain, scale, and schema design. Moreover, TQA-Bench explicitly controls serialized context length (8K–64K tokens) and table formats via relational sampling and symbolic extensions, enabling systematic studies of long-context and serialization effects; MMQA does not target controlled context-length sweeps. To the best of our knowledge, MMQA code and data are not publicly available at the time of writing (the anonymous repository appears to have expired), so we are currently limited to a conceptual comparison rather than an experimental one.

**MTabVQA (Singh et al., 2025).** MTabVQA is a visual multi-tabular QA benchmark designed for vision–language models: models are given images of tables (often across multiple sheets) and must answer questions in the visual modality. In contrast, TQA-Bench operates purely in the text modality: we serialize relational tables into Markdown/CSV/JSON/HTML and feed these token sequences directly to LLMs. As such, MTabVQA and TQA-Bench explore orthogonal dimensions of multi-table reasoning—visual versus text-only interfaces. A quantitative comparison would require extending TQA-Bench with a VLM-based evaluation pipeline, which we view as interesting future work rather than the focus of the present paper.

Table 7: The information of current table QA benchmarks.

| Benchmark | Tabular Data Source | Avg tokens |
|---|---|---|
| Single-Table QA | | |
| WikiTableQuestions (Pasupat & Liang, 2015) | Wikipedia | 1175.05 |
| SQA (Iyyer et al., 2017) | Wikipedia | 554.02 |
| FetaQA (Nan et al., 2022) | Wikipedia | 499.06 |
| HybirdQA (Chen et al., 2020c) | Wikipedia | 601.07 |
| OTT-QA (Chen et al., 2020a) | Wikipedia | 559.61 |
| FinQA (Chen et al., 2021) | FinTabNet (Zheng et al., 2021) | 190.37 |
| AIT-QA (Katsis et al., 2021) | SecGov (Securities & Commission, 2024) | 499.53 |
| Hitab (Cheng et al., 2021) | Wikipedia, Statistical reports | 792.31 |
| TableBench (Wu et al., 2025b) | Wikipedia, FinTabNet, SecGov | 655.46 |
| TATQA (Zhu et al., 2021) | Real-world financial report | 447.91 |
| Multi-Table QA | | |
| Open-WikiTable (Kweon et al., 2023) | Wikipedia | 685.97 |
| Multihiertt (Zhao et al., 2022) | FinTabNet | 1470.48 |
| TSQA (Li et al., 2021) | Chinese high school exams | 410.31 |
| Ours | BIRD (Li et al., 2024b), DataGov (Government, 2024), WorldBank (Group, 2024) | Scale from 8K to 64K |

\* For prior work, Avg tokens is computed per serialized table; for TQA-Bench (ours), per serialized database instance.

## B    BENCHMARK CONSTRUCTION

### B.1    MULTI-TABLE DATA COLLECTION

Many Table QA datasets are based on tables from Wikipedia (Pasupat & Liang, 2015; Iyyer et al., 2017; Kweon et al., 2023; Chen et al., 2020c;a; Cheng et al., 2021; Nan et al., 2022). However, because many LLMs are pre-trained on Wikipedia, its inclusion risks bias and contamination. Moreover, Wikipedia's tables are typically short (only tens of rows) and do not adequately challenge models on comprehensive Table QA tasks. To ensure a rigorous evaluation with complex, unfamiliar tables, we deliberately excluded Wikipedia-derived data. Our data collection instead considers the following sources:

- WORLDBANK.    We incorporated datasets from WorldBank (Group, 2024) to overcome Wikipedia's limitations. WorldBank tables feature extensive rows and columns with simple yet meaningful foreign key relationships that generate actionable insights. Our analysis shows that these datasets often have long-context, multi-table structures—characteristics missing in existing benchmarks. We selected a WorldBank dataset (Dasgupta et al., 2024) that fits our experimental setup and challenges LLMs in realistic, complex scenarios.

- DATAGOV. DataGov (Government, 2024) offers a rich source of real-world tables. Its datasets comprise tables with numerous rows and columns and include basic foreign key relationships that support multi-table reasoning. For our benchmark, we chose two DataGov datasets: the Water Quality Data (of Water Resources, 2024) and Food Facility Inspections (County, 2024). These datasets were sampled and scaled to different context lengths, enabling a wide range of experimental setups.

- BIRD. To complement the above, we included seven databases from BIRD (Li et al., 2024b), a benchmark originally designed for Text2SQL tasks. BIRD databases resemble real-world multi-table environments with complex foreign key relationships; however, many lack referential integrity. Since our sampling requires acyclic, valid foreign key graphs for meaningful queries, we excluded about half of BIRD's databases, narrowing the selection to 20. From these, we carefully chose seven databases that balance semantic richness and manageable complexity, aligning with our benchmark's objectives. The information of the selected databases is shown in Table 8.

Table 8: Structural and query-complexity statistics of the ten TQA-Bench databases.

| Database Name | Source | Table Count | Average #Columns | Average Rows | Total Cells | Join Depth |
|---|---|---|---|---|---|---|
| airline | BIRD | 3 | 10.67 | $2.37 \times 10^5$ | $1.97 \times 10^7$ | 1.79 |
| food_inspection | BIRD | 3 | 8.33 | $2.21 \times 10^4$ | $3.77 \times 10^5$ | 1.64 |
| movie | BIRD | 3 | 9.00 | $2.55 \times 10^3$ | $6.00 \times 10^4$ | 1.21 |
| music_tracker | BIRD | 2 | 5.00 | $1.19 \times 10^5$ | $1.01 \times 10^6$ | 1.5 |
| restaurant | BIRD | 3 | 4.00 | $6.43 \times 10^3$ | $8.66 \times 10^4$ | 1.64 |
| university | BIRD | 6 | 3.33 | $5.34 \times 10^3$ | $1.29 \times 10^5$ | 1.71 |
| cookbook | BIRD | 4 | 9.75 | $2.59 \times 10^3$ | $7.97 \times 10^4$ | 1.43 |
| food_facility_inspections | DataGov | 3 | 13.67 | $1.69 \times 10^5$ | $4.82 \times 10^6$ | 1.64 |
| water_quality | DataGov | 4 | 9.75 | $1.64 \times 10^6$ | $7.01 \times 10^7$ | 1.21 |
| global_biodiversity | WorldBank | 2 | 15.50 | $5.97 \times 10^5$ | $1.85 \times 10^7$ | 1.71 |
| Overall Average | - | 3.3 | 8.36 | $2.83 \times 10^5$ | $1.15 \times 10^7$ | 1.55 |

### B.2    SAMPLING TO VARIATE CONTEXT-LENGTH

Table 8 shows that many selected databases include tables with over $100,000$ rows - far beyond the token limits of mainstream LLMs, which makes direct construction of a multi-table QA dataset impractical. To address this challenge, we develop a sampling method to generate databases of varying context lengths, enabling scalable benchmarking across experimental setups under different computational constraints. The sampling process of the original databases is detailed in Table 9. This sampling process involves two primary steps: (**i**) determine the topological order of the tables based on their foreign key relationships to ensure referential integrity is maintained during sampling; (**ii**) perform the row sampling for each table to create new multi-table database instances from the original databases. These steps ensure that the structural and relational properties of the databases are preserved, even at a reduced scale, allowing for effective benchmarking under various conditions.

**Topological Sort.** The first step in our sampling procedure, as outlined in Algorithm 1, involves determining a topological order among tables to preserve referential integrity during sampling. For-

Table 9: The context length of the serialized sampled database should fall in the range of the minimal and maximal tokens.

| Context Length | Minimum Token Limit | Maximum Token Limit |
|---|---|---|
| 8K | 4000 | 6000 |
| 16K | 8000 | 12000 |
| 32K | 16000 | 24000 |
| 64K | 32000 | 48000 |

mally, for tables within a database, *if table $T_i$ references table $T_j$, then $T_i$ must be sampled prior to $T_j$*. This ordering relies on the assumption that table reference relationships constitute a directed acyclic graph (DAG). While alternative approaches exist for handling cyclic dependencies among tables, such methods complicate precise control over the number of sampled rows per table. Controlling the row count is essential for generating databases with variable context lengths, as it directly influences the scalability and generalizability of our benchmarks across diverse computational settings.

**Row Sampling.** The second step, detailed in Algorithm 2, involves sampling rows from the database while preserving referential integrity. Given a parameter $k$ to determine the sampled number of rows, the algorithm handles tables differently based on their reference dependencies. For tables without incoming references, an ordered sampling of $k$ rows is performed directly. For tables referenced by others, sampling is guided by the topological order of the tables. Rows are selected from the original table that match the referenced column values in the sampled tables, ensuring that all foreign key constraints are respected. To determine the token counts for the sampled databases, we serialize the tables into Markdown format, include table names, and calculate token sizes using a tokenizer. By adjusting the parameter $k$, databases with varying token sizes are generated, approximating the desired context length through a binary search approach. For each target context length, ten database instances were sampled for each original database. The details of the sampled databases are summarized in Table 10.

---

**Algorithm 1** Topological Sort

---

**Require:** $D$: Database instance with tables $\{T_1, ..., T_n\}$ and foreign key dependencies
**Ensure:** A topologically sorted list of tables or an indication of a cyclic dependency
    Initialize an empty list $L \leftarrow \varnothing$
    Create a map $R$, where $R[T_i]$ is the count of incoming references (in-degree) for table $T_i$
    Initialize a set $S \leftarrow \{T_i \mid R[T_i] = 0\}$ containing all tables with zero in-degree
    **while** $S \neq \varnothing$ **do**
      /* Pick any table with zero in-degree */
      Select and remove a table $T \in S$
      Append $T$ to $L$
      **for** each table $U$ referenced by $T$ (i.e., $T \rightarrow U$) **do**
        $R[U] \leftarrow R[U] - 1$
        **if** $R[U] = 0$ **then**
          Add $U$ to $S$
        **end if**
      **end for**
    **end while**
    /* Check for remaining edges indicating a cycle */
    **if** $\exists T_i$ such that $R[T_i] > 0$ **then**
      **return** "Cycle detected"
    **end if**
    **return** $L$

---

### B.3 EVALUATION TASK CATEGORIES

The landscape of table QA benchmarks has evolved substantially over time, reflecting increasingly sophisticated LLM capabilities. Early benchmarks emphasized relatively straightforward tasks, primarily involving direct table lookups and aggregations, which required extracting values or computing basic summaries from tabular data (Pasupat & Liang, 2015; Kweon et al., 2023). As research advanced, benchmarks began to incorporate more intricate tasks demanding numerical reasoning, such

---

**Algorithm 2** Row Sampling with Referential Integrity

---

**Require:** $D$: Database instance
**Require:** $k$: Number of rows to sample from tables without incoming references
**Ensure:** Sampled subset of $D$ maintaining referential integrity
   Initialize an empty list $L \leftarrow \varnothing$
   Compute a topological order $O \leftarrow \text{TOPOLOGYSORT}(D)$
   **for** each table $U \in O$ **do**
      **if** $U$ has no incoming references **then**
         /* Sample $k$ rows from $U$ and preserve row order */
         $T \leftarrow \text{KEEPORDERSAMPLE}(U, k)$
      **else**
         Initialize an empty map $M \leftarrow \varnothing$
         /* Column $A_R$ in $R$ references column $A_U$ in $U$ */
         **for** each reference $R.A_R \rightarrow U.A_U$ **do**
            /* Add referenced values of $A_R$ in $M[A_U]$ */
            $M[A_U] \leftarrow M[A_U] \cup R[A_R]$
         **end for**
         Initialize an empty set $T \leftarrow \varnothing$
         **for** each row $r \in U$ **do**
            **if** any attribute $A$ of $r$ satisfies $r[A] \in M[A]$ **then**
               $T \leftarrow T \cup \{r\}$
            **end if**
         **end for**
      **end if**
      $L \leftarrow L \cup T$
   **end for**
   **return** $L$

---

Table 10: The detailed information on ten databases under four different context lengths.

| Database Name | Average Rows Per Table | | | | Average Token Per Database | | | |
|---|---|---|---|---|---|---|---|---|
| | 8K | 16K | 32K | 64K | 8K | 16K | 32K | 64K |
| airline | 28.60 | 48.30 | 80.07 | 134.17 | $5.07 \times 10^3$ | $9.45 \times 10^3$ | $1.78 \times 10^4$ | $3.42 \times 10^4$ |
| food_inspection | 31.97 | 63.63 | 126.33 | 250.50 | $5.87 \times 10^3$ | $1.16 \times 10^4$ | $2.27 \times 10^4$ | $4.46 \times 10^4$ |
| movie | 23.83 | 47.17 | 92.60 | 180.40 | $5.62 \times 10^3$ | $9.82 \times 10^3$ | $2.11 \times 10^4$ | $4.09 \times 10^4$ |
| music_tracker | 95.90 | 191.95 | 382.70 | 765.25 | $4.92 \times 10^3$ | $9.7 \times 10^3$ | $1.95 \times 10^4$ | $3.89 \times 10^4$ |
| restaurant | 79.50 | 149.40 | 284.07 | 718.93 | $5.16 \times 10^3$ | $9.82 \times 10^3$ | $1.89 \times 10^4$ | $4.85 \times 10^4$ |
| university | 47.28 | 83.07 | 145.32 | 353.55 | $5.33 \times 10^3$ | $9.6 \times 10^3$ | $1.74 \times 10^4$ | $4.49 \times 10^4$ |
| cookbook | 15.82 | 29.82 | 60.62 | 114.28 | $5.88 \times 10^3$ | $1.1 \times 10^4$ | $2.27 \times 10^4$ | $4.28 \times 10^4$ |
| food_facility_inspections | 24.00 | 47.97 | 95.80 | 190.70 | $5.45 \times 10^3$ | $1.06 \times 10^4$ | $2.1 \times 10^4$ | $4.13 \times 10^4$ |
| water_quality | 17.80 | 35.67 | 70.50 | 137.93 | $5.31 \times 10^3$ | $1.04 \times 10^4$ | $2.02 \times 10^4$ | $3.92 \times 10^4$ |
| global_biodiversity | 32.00 | 64.00 | 128.00 | 256.00 | $5.4 \times 10^3$ | $1.06 \times 10^4$ | $2.09 \times 10^4$ | $4.17 \times 10^4$ |
| Overall | 39.67 | 76.10 | 146.60 | 310.17 | $5.4 \times 10^3$ | $1.04 \times 10^4$ | $2.02 \times 10^4$ | $4.17 \times 10^4$ |

as arithmetic operations and understanding numerical relationships, thereby elevating task complexity and sophistication (Chen et al., 2021; Zhao et al., 2022).

Despite these advancements, most current datasets are limited to short-context, single-table scenarios, focusing heavily on analysis within constrained contexts. While they frequently include multi-step arithmetic tasks like addition, subtraction, multiplication, or division (Chen et al., 2021; Zhao et al., 2022), they rarely capture the complexities inherent in long-context, multi-table situations. To address this limitation, our benchmark is explicitly structured around three carefully defined categories—*lookup*, *aggregation*, and *complex calculation*—corresponding to distinct levels of difficulty. This categorization enables a comprehensive assessment across various table QA complexities and scenarios. Illustrative examples for each category appear in Figure 1, and the formal definitions of all subcategories are provided as follows.

Let the database be $\mathcal{D} = (\mathcal{R}, \mathcal{E})$. We use standard relational algebra $\sigma, \pi, \bowtie, \gamma$ for selection, projection, natural join, and group-by aggregation; COUNT, SUM for standard aggregates; and COR for the correlation between 2 selected columns. Given a condition $\Theta$ and a set of attributes $A$, let $\bowtie (A, \Theta)$ denote the minimal join closure that contains all attributes required by $\Theta$ and the output. The formal definitions of each subcategory are as follows:

- *Lookup* tasks are foundational in table-based reasoning. They require the model to locate and extract specific information from tables. We design two tasks in this category:

  ○ *Entity lookup* task retrieves a specific value in the table based on given conditions. Given target attribute $a$ and condition $\Theta$,

  $$\text{EL}(a, \Theta) = \pi_a(\sigma_\Theta(\bowtie(\{a\}, \Theta)))$$

  Our condition $\Theta$ ensures that the final answer is a single item.

  ○ *Top selection* task focuses on identifying key elements or the top entities in a table based on a specific criterion. Given grouping key $g$, a metric $m = \text{AGG}(e)$ (e.g., COUNT, SUM) and condition $\Theta$,

  $$G = \gamma_{g;c:=\text{AGG}(e)}(\sigma_\Theta(\bowtie(\{g, e\}, \Theta)))$$
  $$\text{TS}(g, \Theta, \text{AGG}(e)) = \pi_g(\sigma_{c=\max \pi_c}(G))$$

- *Aggregation* tasks, though conceptually simpler, test an LLM's ability to filter and compute integrated information from the table or the join of multiple tables. We include three aggregation functions in categories:

  ○ *Count* task requires the model to determine the total number of rows or elements satisfying a specific condition. For a specific condition $\Theta$,

  $$\text{CNT}(\Theta) = \text{COUNT}(\sigma_\Theta(\bowtie(\varnothing, \Theta)))$$

  ○ *Sum* task requires the LLM to compute the sum of a specific numerical attribute across the rows that meet certain criteria. For a numerical column $a$ and condition $\Theta$,

  $$\text{SUM}(a, \Theta) = \text{SUM}(\pi_a(\sigma_\Theta(\bowtie(\{a\}, \Theta))))$$

  ○ *Average* task requires the LLM to calculate the mean of a numerical column for rows matching conditions. For a numerical column $a$ and condition $\Theta$,

  $$\text{AVG}(a, \Theta) = \frac{\text{SUM}(a, \Theta)}{\text{CNT}(\Theta)}$$

- *Complex calculation* tasks evaluate advanced reasoning capabilities, focusing on more intricate operations. We categorize these into two subcategories:

  ○ *Composite comparison* task requires the LLM to compare the difference between two values, which may either be directly available in the table or derived through intermediate calculations. For a comparison expression $e$ (e.g., $e = \text{ARR\_DELAY-DEP\_DELAY}$), a metric $m = \text{AGG}(e)$ and condition $\Theta$,

  $$\text{CC}(\text{AGG}(e), \Theta) = \text{AGG}(\pi_a(\sigma_\Theta(\bowtie(\{e\}, \Theta))))$$

  ○ *Correlation* task requires the LLM to compute the statistical relationship between two numeric columns. For numeric columns $a$, $b$ and condition $\Theta$, the answer is the Pearson correlation coefficient over the rows satisfying $\Theta$:

  $$\text{COR}(a, b, \Theta) = \frac{\sum_{i \in I_\Theta}(a_i - \bar{a})(b_i - \bar{b})}{\sqrt{\sum_{i \in I_\Theta}(a_i - \bar{a})^2}\sqrt{\sum_{i \in I_\Theta}(b_i - \bar{b})^2}},$$

  where $I_\Theta$ indexes rows satisfying $\Theta$, and $\bar{a}$, $\bar{b}$ are the corresponding sample means.

Unlike existing Table QA benchmarks, our tasks are clearly categorized across multiple tables, enabling targeted question creation and systematic performance evaluation. By organizing tasks hierarchically - from simple lookups to complex calculations - our benchmark compares model performance across a spectrum of challenges. It also emphasizes scalability and multi-table contexts, filling critical gaps in current datasets while maintaining practical reasoning depth. This structure enhances evaluation robustness and promotes the model's capabilities of handling intricate situations.

### B.4 QUESTION GENERATION BY SYMBOLIC EXTENSION

Inspired by the GSM-Symbolic framework (Mirzadeh et al., 2024), we adopt symbolic extension in our benchmark to generate a larger set of high-quality evaluation questions. By combining symbolic extension with sampling, we create diverse and meaningful queries, enhancing the benchmark's robustness for LLM evaluation.

**Symbolic Question Generation.** Our symbolic extension is divided into two principal components: template question design, and the generation of questions and solutions, as depicted in Figure 1. Template questions are crafted with placeholder variables instead of fixed values, enabling dynamic content generation. These variables are subsequently instantiated, and the correct answers are computed using Python implementation. This methodology facilitates the creation of multiple question instances from a single template, thereby enhancing the benchmark's versatility and scalability. To enable a more effective evaluation, we employ multiple-choice questions (MCQs) rather than relying solely on traditional metrics such as exact match, BLEU, or F1 scores. These conventional metrics can fall short of accurately assessing the reasoning capabilities of LLMs. MCQs offer a more direct method to evaluate understanding and reasoning by providing discrete, comparable options (Balepur & Rudinger, 2024). Moreover, many of our items involve multi-step analytical computation, where current LLMs often struggle to produce exact numeric answers. Framing these items as MCQs yields unambiguous scoring, reduces spurious partial matches, and aligns with established practice in STEM and mathematics benchmarks (Hendrycks et al., 2020; Wang et al., 2024; Amini et al., 2019). For each database, we manually design two template questions per subcategory, each paired with a corresponding Python code solution. This approach yields a total of 140 template questions across all databases and subcategories. To populate the benchmark with diverse instances, we leverage the ten database instances created for each database and context length. Using the symbolic extension, we generate ten question instances for each template question. An overview of the total number of benchmark instances is provided in Table 11, illustrating the extensive scale and scope of our benchmark. More examples of the generation process for the Airline database in Figure 1 are provided in Figure 5.

Table 11: An overview of our benchmark instances.

| Context Length | Database Instances | Question Instances |
|---|---|---|
| 8K | 100 | 14000 |
| 16K | 100 | 14000 |
| 32K | 100 | 14000 |
| 64K | 100 | 14000 |
| Total | 400 | 56000 |

**Wrong Choice Generation**. To create incorrect options for the MCQs, we use a rule-based approach. For *entity lookup* and *top selection* tasks, we randomly select different cells from the same column to generate error choices. For the rest tasks, which require numerical answers, we produce three error options by multiplying the correct answer by 0.25, 2.0, and 3.0. While this method is simple, our experiments show that these questions remain challenging for LLMs, especially due to the design consideration that our tasks require reasoning over long contexts and multiple tables.

## C SELECTED MODEL DETAILS

We list the LLMs included in the comprehensive evaluation below:

- GPT (OpenAI, 2024c;b; 2025b; 2024d): we evaluate GPT-4O-MINI, GPT-4O, GPT-O1-MINI and GPT-O3-MINI from OpenAI as the state-of-the-art close-source models. All of them are tailored for conversational AI and reasoning tasks, supporting context up to 128K tokens.
- QWEN2.5 (Team, 2024b): we select the 3B, 7B, 14B, 72B versions of QWEN2.5 Instruct model and a 7B Coder model for evaluation. All of them support long context up to 128K tokens.
- QWQ (Team, 2024c): it is a reasoning model that was developed by the Qwen team. It supports up to 32K tokens.
- LLAMA3.1 (Dubey et al., 2024b): we include the 8B and 70B LLAMA3.1 Instruct. Both of them support a context length up to 128K tokens.

(a) Top Selection: Which airport lands most flights start from `ORIGIN`?

```
1 ORIGIN = self.Airlines['ORIGIN'].sample(1).iloc[0]
2 origin_description = self.Airports[self.Airports['Code'] ==
      ORIGIN]['Description'].iloc[0]
3 filted = self.Airlines[self.Airlines['ORIGIN'] == ORIGIN]
4 max_count = filted['DEST'].value_counts()
5 max_val = max_count.max()
6 lands_airport = max_count[max_count == max_val].index
7 dest_description =
      self.Airports[self.Airports['Code'].isin(lands_airport)]['Description'].to_list()
```

(b) Count: How many airlines land in `DEST`?

```
1 DEST = self.Airlines['DEST'].sample(1).iloc[0]
2 dest_description = self.Airports[self.Airports['Code'] ==
      DEST]['Description'].iloc[0]
3 filted = self.Airlines[self.Airlines['DEST'] == DEST]
4 land_airline = len(filted)
```

(c) Average: What is the average flight delay (ARR_DELAY) that land in `DEST`?

```
1 DEST = self.Airlines['DEST'].sample(1).iloc[0]
2 dest_description = self.Airports[self.Airports['Code'] ==
      DEST]['Description'].iloc[0]
3 filted = self.Airlines[self.Airlines['DEST'] == DEST]
4 avg = filted['ARR_DELAY'].mean()
```

Figure 5: Additional QA generation examples.

- BAICHUAN2 (Yang et al., 2023): we include BAICHUAN2 7B chat model and 13B chat model. Both of them support long context up to 192K tokens.

- GEMMA2 (Team, 2024a): we select the 2B, 9B, and 27B versions of GEMMA2 Instruct model for evaluation. All of them only support a context length up to 8K tokens.

- GLM-4 (GLM et al., 2024): we evaluate GLM-4-9B-CHAT on the benchmark. The model is a chat model with a context length of 128K tokens.

- MISTRAL (Jiang et al., 2023): we evaluate MISTRAL-NEMO-INSTRUCT and MISTRAL-7B-INSTRUCT on the benchmark. Both of them are instruct models. MISTRAL-NEMO-INSTRUCT is trained with 12.2B parameters and supports up to 128K context window. MISTRAL-7B-INSTRUCT is trained with 7B parameters and supports up to 32k tokens.

- VICUNA (Chiang et al., 2023): we select VICUNA-7B-V1.5-16K and VICUNA-13B-V1.5-16K to evaluate. As the name suggests, VICUNA-7B-V1.5-16K is a chat model trained with 7B parameters and supports up to 16k tokens. VICUNA-13B-V1.5-16K is a chat model trained with 13B parameters and supports up to 16k tokens.

- TABLELLAMA (Zhang et al., 2024a): The TABLELLAMA model is fine-tuned on the TableInstruct dataset using LongLoRA so that it is specialized in table-based tasks. The size of the model is 7B but it only supports a context length up to 8k tokens.

- TABLEGPT2 (Su et al., 2024): The TABLEGPT2 is derived from the QWEN2.5 architecture and specialized in analyzing tabular data. However, it is trained mostly on Chinese corpora and may not support other languages well. The model size is 7B, and it supports up to 128K tokens as input.

- ARCTIC-TEXT2SQL-R1 (Yao et al., 2025): The ARCTIC-TEXT2SQL-R1 model is trained on the QWEN2.5 series and specialized in Text2SQL tasks. It achieves state-of-the-art performance on the BIRD (Li et al., 2024b) benchmark. The size of the model ranges from 7B to 32B.

- DEEPSEEK-V3 (DeepSeek-AI, 2024): we evaluate DEEPSEEK-V3 on the benchmark. Unlike dense architecture, this model adopts a MoE architecture. It is trained with 671B parameters, and it supports a context length of 128K tokens.

- DEEPSEEK-R1 (Guo et al., 2025): The DEEPSEEK-R1 is a widely recognized reasoning model that enhances its reasoning capabilities through reinforcement learning. It includes a full version, post-trained from DEEPSEEK-V3, along with several distilled variants derived from the full

model. In our study, we select the full DEEPSEEK-R1-671B model and two distilled versions: DEEPSEEK-R1-DISTILL-QWEN-7B and DEEPSEEK-R1-DISTILL-QWEN-14B.

# D EXPERIMENT DETAILS

## D.1 EXPERIMENT 1

**Dataset setup.** We conduct this experiment on the `airline` database at the 8K context length, using all ten sampled database instances and 1,400 generated questions. For *single-table* evaluation, for each question we materialize a denormalized table by pre-joining exactly those base tables referenced by the question. This produces a question-specific single table that preserves the same answer as the original multi-table query. Each question is then evaluated under both settings: (i) the original *multi-table* schema, and (ii) the corresponding *single-table* (pre-joined) version, enabling a controlled comparison across the two contexts.

**Evaluated LLMs.** GLM-4-9B-CHAT, QWEN2.5-7B-INSTRUCT, LLAMA3.1-8B-INSTRUCT, DEEPSEEK-R1-DISTILL-QWEN-7B.

**LLM selection rationale.** This experiment studies how single- vs. multi-table structural settings affect accuracy under a fixed context budget. We therefore choose a small but diverse set of mid-sized models: a chat-oriented LLM (GLM-4-9B-CHAT), two instruction-tuned LLMs from different families (QWEN2.5-7B-INSTRUCT and LLAMA3.1-8B-INSTRUCT), and a reasoning-oriented model (DEEPSEEK-R1-DISTILL-QWEN-7B). This configuration lets us probe whether the single–multi gap is consistent across model categories while keeping the cost of running all context-length and sampling variants manageable.

## D.2 EXPERIMENT 2

**Dataset setup.** For each database and each context length, we sample five database instances. For each instance, we select one question instance for each of the 14 question templates, yielding 50 database instances and 700 questions per context length. This protocol is used to compare performance across formats and scales while keeping the per-instance question diversity fixed.

**Evaluated LLMs.** QWEN2.5-7B-INSTRUCT, QWEN2.5-CODER-7B-INSTRUCT, LLAMA3.1-8B-INSTRUCT.

**LLM selection rationale.** This experiment isolates how different serialization formats (Markdown, CSV, JSON, HTML) interact with model pre-training. To control for model-side factors, we keep two general-purpose instruction-tuned baselines from Experiment 1 (QWEN2.5-7B-INSTRUCT and LLAMA3.1-8B-INSTRUCT) and additionally include a coder-oriented variant (QWEN2.5-CODER-7B-INSTRUCT). This allows us to contrast general vs. code-specialized instruction tuning while reusing the same 7B–8B scale to stay within our computation budget.

**Model-Specific Observations.** Our analysis revealed that coder LLMs often outperform their base counterparts, with the extent of improvement depending on the serialization format. In the CSV format, the QWEN2.5-CODER-7B-INSTRUCT outperformed QWEN2.5-7B-INSTRUCT across all scales. For other formats, performance improvements were observed in specific scales.

## D.3 EXPERIMENT 3

**Dataset setup.** We adopt the same protocol as Experiment 2 (five instances per database and context length; 14 templates per instance), and evaluate at 8K, 16K, 32K, and 64K context lengths. This setting allows us to study scale effects and long-context robustness across open- and closed-source models.

**Evaluated LLMs.** We evaluate LLMs on following two categories:

*Open-source:* QWEN2.5, LLAMA3.1, BAICHUAN2, GLM-4, MISTRAL, DEEPSEEK-V3, DEEPSEEK-R1, TABLEGPT2; additionally, GEMMA2, TABLELLAMA, and VICUNA are evaluated up to their respective context-length limits.

*Closed-source:* GPT-4O, GPT-4O-MINI, GPT-O1-MINI, GPT-O3-MINI via the OpenAI API.

**LLM selection rationale.** This experiment constitutes our main benchmark evaluation. After fixing Markdown as the default serialization format based on Experiment 2, we run all 28 LLMs listed in Appendix C on TQA-Bench across all four context-length settings. These results provide the global comparison that subsequent in-depth analyses (Experiments 4–5) build upon.

**Model-Specific Observations.** We enumerate our interesting model-specific observations based on LLM categories:

*Chat LLM Performance.* A manual inspection using the airline database revealed distinct failure patterns for different models. For instance, larger versions of BAICHUAN2 frequently exhibited repetitive output (e.g., "`-338.166666666666...`" repeated many times), whereas smaller versions, while often failing to follow instructions, still managed to produce readable and structured responses. For the VICUNA series, smaller models occasionally produced multiple-choice outputs (e.g., "`C/D`"), where our regex could possibly still capture one valid option, whereas larger versions tended to produce verbose answer or "`None of the above`".

*Instruct LLM Performance.* Although instruct models generally outperform chat models, **two exceptions stand out:** MISTRAL and QWEN2.5. Despite being instruction-tuned, MISTRAL frequently outputs "`None of the above`", especially with long contexts, indicating diminished instruction adherence. For QWEN2.5, the largest version does not show a significant improvement in overall accuracy. Our manual inspection of its generated outputs reveals that its tendency to produce verbose analyses often pushes outputs beyond the token limit, leaving no space for providing a valid final answer.

*Domain-Specific Tabular LLMs.* Although designed for table-based tasks, domain-specific models such as TABLELLAMA and TABLEGPT2 do not meet performance expectations in our benchmark. We hypothesize that overspecialization may narrow their adaptability: TABLELLAMA fails to follow the required output format, and TABLEGPT2, while format-compliant, delivers only average results. This behavior suggests that their continuous pre-training may not have adequately represented the full range of question types or formats used in our evaluation. Furthermore, these models may not optimally balance between relational data generation and the flexibility required for general QA tasks, indicating a potential misalignment between training objectives and the evaluation criteria introduced by our benchmark.

*Reasoning LLM Performance.* Distilled models consistently underperform their original counterparts, with distillation particularly harming smaller models' ability to handle long contexts. For instance, QWEN2.5-7B-INSTRUCT can follow instructions at the 64K scale, whereas its distilled variant fails. Interestingly, QWQ-32B-PREVIEW shows anomalous behavior, frequently producing repetitive and meaningless tokens—a pattern also observed in DEEPSEEK-R1-DISTILL-QWEN-7B at the 64K scale.

**Task-Specific Observations under Context Length.** The effect of context length varies across different categories of questions:

- *Lookup tasks* decline relatively slowly, as they often require retrieving only a single or few items, which remains manageable even with longer contexts.

- *Aggregation tasks* suffer sharper declines. Notably, models perform better on *average* questions than on *sum* questions, since estimating an approximate average is more intuitive, whereas summation requires precise computation.

- For *complex calculations*, the impact depends on the subcategory. Composite comparison tasks retain relatively stable performance, while correlation tasks show the steepest declines, likely because they demand both complex numerical computation and logical reasoning.

### D.4   EXPERIMENT 4

**Dataset setup.** We reuse the same dataset as Experiment 1: the `airline` database at the 8K context length, the same ten sampled database instances, and the same $1,400$ generated question instances. For analysis granularity, we organize questions into *batches* at the instance level: for each database instance, we create 10 batches; each batch contains 14 questions, one from each of the 14 templates (thus $10 \times 14 = 140$ questions per instance and $10 \times 10 \times 14 = 1,400$ in total). Batches are indexed

by the pair (database-instance index, batch index) to support controlled cross-batch comparisons under a fixed schema and data sample.

**Evaluated LLMs.** GLM-4-8B-CHAT, QWEN2.5-7B-INSTRUCT, LLAMA3.1-8B-INSTRUCT, GPT-4O-MINI, DEEPSEEK-R1.

**LLM selection rationale.** This experiment provides in-depth analyses (e.g., sampling and symbolic extension) on top of the global results in Experiment 3. To keep the analysis readable while still covering the main model axes, we select a small but representative subset of strong LLMs: GLM-4-9B-CHAT, QWEN2.5-7B-INSTRUCT, LLAMA3.1-8B-INSTRUCT, GPT-4O-MINI, and DEEPSEEK-R1. This subset spans open- vs. closed-source models and chat-, instruction-, and reasoning-oriented paradigms.

**Model-Specific Observations.** While overall trends are consistent, some models exhibit distinct behaviors. For example, a batch in the lower-left corner of the heatmap appears easier for QWEN2.5-7B-INSTRUCT, which achieves high accuracy, but presents average difficulty for other models. In comparative tests, all models except GPT-4O-MINI showed similar results between broad and sensitive airline evaluations. Furthermore, most models perform better when evaluated across all databases than on the airline database alone, indicating that "airline" contains relatively challenging questions. Interestingly, DEEPSEEK-R1 displays the opposite trend, performing better on the airline database than across all databases, suggesting that these instances are comparatively simpler for this model.

## D.5  EXPERIMENT 5

**Dataset setup.** We evaluate multiple context lengths using the same $2,800$ questions and database instances from Experiment 3. Under the Text2SQL setting, models are provided with database schemas and prompted to produce executable SQL queries using an instruction adapted from ARCTIC-TEXT2SQL-R1 (Yao et al., 2025) (see Appendix §E). Execution correctness is measured by running the generated SQL against the corresponding database instance.

**Evaluated LLMs.** GLM-4-9B-CHAT, QWEN2.5-7B-INSTRUCT, LLAMA3.1-8B-INSTRUCT, ARCTIC-TEXT2SQL-R1, GPT-4O-MINI, DEEPSEEK-R1. Among these, ARCTIC-TEXT2SQL-R1 is a representative Text2SQL-specialized LLM that ranks highly on BIRD-Bench (BIRD-bench, 2025).

**LLM selection rationale.** This experiment compares direct end-to-end prompting with an LLM-based Text2SQL pipeline. We reuse the representative subset from Experiment 4, replacing QWEN2.5-7B-INSTRUCT with the coder-optimized QWEN2.5-CODER-7B-INSTRUCT, and additionally include the Text2SQL-specialized model ARCTIC-TEXT2SQL-R1. This configuration reflects typical Text2SQL practice (coder-style models plus a dedicated Text2SQL baseline) while keeping the setup consistent with our earlier analyses.

## E  PROMPTS IN THE EVALUATION

We attach the prompt used in our benchmarks below:

**LLM Prompts in Table QA.** We design the prompts to be as simple and universally applicable as possible for end-to-end TableQA evaluations, while supporting various methods of encoding tables as input. The following prompt is used in Experiments 1-4, and the direct LLM prompt in Experiment 5.

---

**Bascic Prompt for Experiment 1 to 4.**

Please carefully analyze and answer the following single choice question step by step.

## Database: {database name}

### Table: {table name 0}

`{table 0 in markdown/csv/html/json}`

---

> **Table: {table name 1}**
>
> `{table 1 in markdown/csv/html/json}`
>
>
> ...
>
> **Question:**
> `{question}`
>
> A) `{choice A}`
> B) `{choice B}`
> C) `{choice C}`
> D) `{choice D}`
>
> This question has only one correct answer. Please break down the question, evaluate each option, and explain why it is correct or incorrect. Conclude with your final choice on a new line formatted as `Answer: A/B/C/D`.

**Text2SQL Prompt.** The following prompt template is used in Experiment 5 to guide LLMs to generate SQL, i.e., LLM based-Text2SQL. The prompt is based on the original prompt from the report of ARCTIC-TEXT2SQL-R1 (Yao et al., 2025), but relaxes strict formatting requirements to allow the evaluation of a broader range of models.

> **Text2SQL Prompt for Experiment 5.**
>
> You are a data science expert. Below, you are provided with a database schema and a natural language question. Your task is to understand the schema and generate a valid SQL query to answer the question.
>
> **Database Engine:**
> SQLite
>
> **Database Schema:**
> `{schema}`
> This schema describes the database's structure, including tables, columns, primary keys, foreign keys, and any relevant relationships or constraints.
>
> **Question:**
> `{question}`
>
> **Instructions:**
> - Make sure you only output the information that is asked in the question. If the question asks for a specific column, make sure to only include that column in the `SELECT` clause, nothing more.
> - The generated query should return all of the information asked in the question without any missing or extra information.
> - Before generating the final SQL query, please think through the steps of how to write the query.
>
> **Output Format:**
> Please provide a detailed chain-of-thought reasoning process. Ensure that your SQL query follows the correct syntax and is formatted as follows:
>
> ```sql
> -- Your SQL query here
> ```

## F  FRONTIER MODEL AND AGENT TESTING

We additionally evaluate a strong frontier model, GPT-5.1 (OpenAI, 2025a), on the 64K setting. The results are shown in Table 12.

Table 12: Performance of GPT-5.1 on TQA-Bench (64K setting).

| EL | TS | CNT | SUM | AVG | CC | COR | Average |
|---|---|---|---|---|---|---|---|
| 94.85 | 93.81 | 58.16 | 57.29 | 72.63 | 90.82 | 53.54 | 74.41 |

We observe that GPT-5.1 obtains only 53.54% on the 64K COR task, a result comparable to earlier models such as GPT-O3-MINI. In other words, despite approximately ten months of rapid model development since DeepSeek-R1's release (Jan 2025), frontier LLMs still struggle significantly on the long-context COR setting.

This lack of improvement suggests that (i) the 64K analytical tasks remain far from being solved, and (ii) current frontier models still exhibit clear weaknesses in long-context multi-table reasoning. Therefore, TQA-Bench is not saturated and retains meaningful long-term utility for the community as a target for evaluating future progress.

We further clarify our position on current agentic AI workflows. The agentic ecosystem today contains many heterogeneous workflow designs. As recent work (e.g., FDABench) highlights, current research proposes diverse agent workflows but lacks standardized implementations. FDABench attempts to standardize four representative workflow patterns, yet it still concludes that a universally accepted agent workflow does not exist (Wang et al., 2025). Hence, instead of broadly surveying agent workflows, our work focuses on a more fundamental and actionable experimental question: In a complex workflow, which implementation of the table agent should be used to best improve the full agent system? (i) direct prompting, and (ii) with a code interpreter (Python engine).

For the COR task, we compare GPT-5.1 (64K) under direct prompting versus code interpreter, and observe that replacing direct prompting with code interpreter yields substantial improvements. The model achieves 60.53% accuracy, which is higher than 53.54%.

## G ERROR ANALYSIS

### G.1 EFFECT OF THE MULTIPLE-CHOICE ANSWER FORMAT

One concern about our evaluation protocol is that the single-choice A/B/C/D format, together with prompts that insist on "output a single letter," might punish otherwise correct solutions that are phrased in a different way (e.g., a correct numeric answer without the final option letter). To examine whether this is a dominant failure mode in practice, we manually inspected 50 mispredictions made by GPT-4O at the 8k scale in the multiple-choice setting.

For all 50 examined cases, we found that the model's reasoning or final numerical conclusion was incorrect relative to the ground-truth answer. In 46 out of 50 cases, the model produced a well-formed single-letter choice ("A"–"D") that was fully consistent with its (incorrect) reasoning. In the remaining 4 cases, the model gave answers such as `Answer: None`, i.e., it confidently stated that no option was correct, which contradicts our dataset construction where exactly one option is guaranteed to be correct. These were consistently marked as incorrect.

Crucially, in this sample we did not observe instances where the model arrived at the correct numeric value but was scored as incorrect solely because it failed to output the desired letter format. This suggests that, at least for a strong model such as GPT-4O at 8k, the main source of errors under our evaluation protocol is the underlying analytical reasoning, rather than the answer-format restriction itself. We acknowledge that more fine-grained partial-credit schemes (e.g., parsing and evaluating intermediate numeric outputs) could be explored in future work, but we adopt the single-letter multiple-choice design here to enable scalable, objective, and unambiguous automatic evaluation across tens of thousands of questions.

This multiple-choice, exact-match evaluation protocol is consistent with many widely used LLM benchmarks (Hendrycks et al., 2020; Wang et al., 2024; Amini et al., 2019), where models are also scored by exact match on the selected option rather than by partial credit.

## G.2 TEXT2SQL: QUALITATIVE ERROR PATTERNS ON COMPOSITE QUERIES

To better understand why top models struggle on correlation and composite analytical tasks in the Text2SQL setting, we manually analyzed 50 mispredicted queries produced by DEEPSEEK-R1 at the 8k scale. These queries are dominated by correlation and multi-step arithmetic questions. We identify three recurring error patterns:

- **Non-executable or non-SQL output (approximately** $26$–$52\%$**).** The model sometimes produces a mathematical formula instead of an executable SQL query, especially for COR questions. For example, it may output `correlation = (n * sum_xy - sum_x * sum_y) / SQRT((n * sum_x2 - sum_x^2) * (n * sum_y2 - sum_y^2))` as the "final SQL query". While this expression is mathematically meaningful, it cannot be executed by the SQL engine and therefore fails under our evaluation protocol.

- **Wrong solution step or task formulation (approximately** $16$–$32\%$**).** In these cases, the model produces syntactically valid SQL, but the query does not implement the correct computational step required by the question. For instance, for the question "How many budgets is *Remember the Titans* higher than *X-Men Origins: Wolverine*?", one generated query is:

```
SELECT
    CASE
        WHEN (SELECT Budget FROM movie WHERE Title = 'Remember the Titans') >
                (SELECT Budget FROM movie WHERE Title = 'X-Men Origins: Wolverine')
        THEN 1
        ELSE 0
    END AS Count;
```

This query only checks whether one budget is larger than the other and returns a binary indicator (1 or 0), instead of computing the numeric difference between the two budgets as requested.

- **Correct high-level plan but mis-specified expression (approximately** $8$–$16\%$**).** Here the model outlines a reasonable multi-step strategy in SQL, but small expression-level choices lead to a mismatch with the intended semantics. For example, consider the question "What is the average total fly time (ARR_TIME - DEP_TIME) of United Air Lines Inc.: UA?". One generated query is:

```
SELECT AVG(
    CASE
        WHEN (ARR_TIME / 100 * 60 + ARR_TIME % 100) >=
                (DEP_TIME / 100 * 60 + DEP_TIME % 100)
        THEN (ARR_TIME / 100 * 60 + ARR_TIME % 100) -
                (DEP_TIME / 100 * 60 + DEP_TIME % 100)
        ELSE (ARR_TIME / 100 * 60 + ARR_TIME % 100 + 1440) -
                (DEP_TIME / 100 * 60 + DEP_TIME % 100)
    END) AS average_fly_time
FROM Airlines
JOIN "Air Carriers"
  ON Airlines.OP_CARRIER_AIRLINE_ID = "Air Carriers".Code
WHERE "Air Carriers".Description = 'United Air Lines Inc.: UA'
  AND Airlines.CANCELLED = 0
  AND Airlines.DEP_TIME IS NOT NULL
  AND Airlines.ARR_TIME IS NOT NULL;
```

This query attempts to interpret times in HHMM format and converts them to minutes using `t/100*60 + t%100`, while also handling overnight flights via a 1440-minute wrap-around. Although this logic is intricate, it does not correspond to the ground-truth interpretation used in our benchmark, where the task is defined clearly as the direct difference `ARR_TIME - DEP_TIME`. Thus, a seemingly "overly careful" expression actually moves the answer away from the target computation.

Overall, these qualitative patterns support our main claim that the bottleneck for Text2SQL models on correlation and composite tasks lies in composing the right sequence of SQL operations and expressions, rather than merely recalling individual operators. Models can often identify relevant tables and columns, but they frequently fail to (i) express statistical formulas in valid SQL, (ii) select the correct computational step for the question (e.g., difference vs. comparison), or (iii) align their detailed time and arithmetic handling with the benchmark's problem formulation.

## H STATISTICAL ROBUSTNESS OF FORMAT COMPARISON

In Section 4.2, Table 4 compares four serialization formats and shows that Markdown typically achieves higher accuracies than the alternatives across models and context lengths. Since HTML and JSON do not offer a better accuracy–length trade-off under our settings, the practically relevant choice is between Markdown and CSV: Markdown tends to yield higher accuracy, while CSV is more compact in terms of serialization length. Tables 13 and 14 therefore focus on a direct Markdown–CSV comparison.

Table 13: McNemar $\chi^2$ tests comparing Markdown vs. CSV for three instruction-tuned models on the 8k–64k multi-table QA tasks.

| Model | Context | $\chi^2$ | $p$-value |
|---|---|---|---|
| Qwen2.5-7B-Instruct | 8k | 19.79 | $8.65 \times 10^{-6}$ |
| | 16k | 42.00 | $9.11 \times 10^{-11}$ |
| | 32k | 13.32 | $2.63 \times 10^{-3}$ |
| | 64k | 0.11 | 0.74 |
| Qwen2.5-Coder-7B-Instruct | 8k | 0.019 | 0.89 |
| | 16k | 3.54 | 0.060 |
| | 32k | 22.44 | $2.17 \times 10^{-6}$ |
| | 64k | 0.11 | 0.74 |
| Llama3.1-8B-Instruct | 8k | 1.14 | 0.29 |
| | 16k | 4.72 | 0.030 |
| | 32k | 1.31 | 0.25 |
| | 64k | 0.0 | 1.0 |

Table 14: Average accuracy (%) and 95% binomial confidence intervals for Markdown (MD) and CSV formats across models and context scales.

| Model | Fromat | 8k | 16k | 32k | 64k |
|---|---|---|---|---|---|
| Qwen2.5-7B-Instruct | MD | 49.86(47.97-51.75) | 50.86(48.97-52.75) | 41.57(39.71-43.43) | 33.43(31.65-35.21) |
| | CSV | 40.57(38.71-42.43) | 36.86(35.04-38.68) | 33.71(31.92-35.50) | 32.57(30.80-34.34) |
| Qwen2.5-Coder-7B-Instruct | MD | 48.0(46.11-49.89) | 47.14(45.25-49.03) | 44.86(42.98-46.74) | 36.86(35.04-38.68) |
| | CSV | 47.57(45.68-49.46) | 43.0(41.13-44.87) | 34.43(32.63-36.23) | 36.0(34.19-37.81) |
| Llama3.1-8B-Instruct | MD | 46.57(44.68-48.46) | 43.0(41.13-44.87) | 35.43(33.62-37.24) | 35.86(34.05-37.67) |
| | CSV | 44.29(42.41-46.17) | 38.71(36.87-40.55) | 33.0(31.22-34.78) | 35.71(33.90-37.52) |

Table 13 reports McNemar $\chi^2$ statistics and $p$-values for paired per-question correctness between Markdown and CSV for three instruction-tuned models and four context scales. Each question instance is treated as a paired binary outcome (correct / incorrect) for the two formats, and the test is conducted with one degree of freedom. For QWEN2.5-7B-INSTRUCT, the Markdown–CSV gaps are highly significant at 8k, 16k, and 32k ($p \ll 0.01$), but not at 64k, where the two formats have very similar accuracies. QWEN2.5-CODER-7B-INSTRUCT shows a strong and significant Markdown advantage at 32k. In all cases, the large $\chi^2$ values and very small $p$-values coincide with the largest raw accuracy gaps in Table 4.

Table 14 complements this analysis by reporting point estimates and 95% binomial confidence intervals for the average accuracy of Markdown and CSV at each context scale. The intervals are fairly tight (typically within $\pm 1$–2.5 percentage points). Whenever McNemar's test indicates a significant Markdown–CSV difference, the corresponding intervals for Markdown and CSV do not overlap, while in the remaining conditions the intervals overlap substantially, reflecting that the two formats are statistically indistinguishable there. Taken together, these diagnostics confirm that the main

Markdown gains highlighted in Section 4.2 are statistically robust in several core model–context configurations, while also clarifying that Markdown does not uniformly dominate CSV across all settings and that CSV remains competitive whenever the accuracy gaps are small. Considering both accuracy and serialization efficiency—and the severe token overhead of HTML—these results support our choice of Markdown as a strong default serialization format for TQA-Bench.

## I  THE USE OF LLMs IN WRITING

We used LLM, namely OPENAI-GPT5, to polish the writing of this manuscript. No other generative AI functionality is used in the writing of this submission.

