# OpenReview forum: "TQA-Bench: Evaluating LLMs for Multi-Table Question Answering"
_ICLR.cc/2026/Conference — ICLR 2026 Conference Withdrawn Submission_

### Official Review · Reviewer_sY5W · 2025-10-26

**Soundness:** 2
**Presentation:** 3
**Contribution:** 3
**Rating:** 4
**Confidence:** 4

**Summary:**

The paper introduces TQA-Bench, a new benchmark for evaluating LLMs on multi-table question answering (QA) tasks. The authors argue that existing table QA benchmarks largely focus on single-table contexts and fail to capture the relational complexity of real-world datasets.

TQA-Bench is built from three data sources (World Bank, Data.gov, and BIRD) and supports variable context lengths (8K–64K tokens). The queries are generated following the common approach of generating some templates and filling them (referred to as symbolic extension). The authors evaluate a wide range of open- and closed-source LLMs (2B–671B parameters), exploring the effects of table serialization formats (Markdown, CSV, JSON, HTML), table size, and comparison between direct prompting and Text2SQL approaches. Results show that:

- Multi-table QA is significantly more challenging than single-table QA.

 - Markdown serialization performs best across formats.

 - Model performance degrades with longer contexts.

 - Text-to-SQL approaches scale better with context length, though they struggle on analytical queries.

**Strengths:**

* **Filling a gap in table reasoning**: Aims to addresses a gap in benchmarks for multi-table QA, a domain of increasing relevance for LLM-based data management.

* **Comprehensive evaluation**: Includes an extensive set of LLMs, formats, and context sizes, producing useful empirical insights.

* **Symbolic extension in question generation**: The idea of augmenting quetion templates with symbolic augmentations allows automatic creation of benchmark questions and and assessing reasoning depth.

* **Scalable dataset design**: The sampling mechanism for controlling serialized length (8K–64K) is well-motivated for long-context model evaluation.

**Weaknesses:**

* **Missing related work on multi-table QA**:
  The paper overlooks directly relevant and recent multi-table QA datasets such as

  * Wu et al., Evaluating LLMs with Multi-Table Multi-Hop Complex Questions. ICLR 2025.

  * Singh et al., MTabVQA: Evaluating Multi-Tabular Reasoning of Language Models in Visual Space. arXiv preprint arXiv:2506.11684 (2025)

  Both address similar goals and should be discussed and compared experimentally or at least conceptually. Without this, the novelty claim is weakened.

* **Dataset scope and representativeness**:
  The included databases (especially those from BIRD) are highly structured and relational, which may not reflect the semi-structured nature of many real-world tables commonly used in reasoning tasks. This limits the generality of conclusions and may bias the benchmark toward text-to-SQL-style reasoning. A clearer discussion contrasting structured (relational) vs. semi-structured table QA would be valuable.

* **Lack of dataset analysis**:
  The paper spends much of its space on model results (Tables 3–5) but omits essential dataset diagnostics, such as the number of databases, query types per complexity class, or distribution of join depth. Such analysis is necessary to assess the diversity and difficulty of TQA-Bench.

* **Formatting and presentation**:
  While generally clear, several sections are overly detailed (e.g., exhaustive model tables) at the expense of key conceptual comparisons. The narrative could be improved by focusing more on dataset design choices and benchmark positioning rather than numerical results alone.

* **Unclear size compression explanation**:
  The token count for Markdown, CSV, and JSON in Table 3 decreases from 8K to smaller values (5.4k, 3.7k, 5.75k tokens), which seems counterintuitive. Clarification or an example of the serialization process would help ensure reproducibility.

**Questions:**

1. How does TQA-Bench compare quantitatively to MMQA and MTabVQA in size, complexity, and domain coverage?

2. Are any of the databases semi-structured (e.g., with missing relations, nested tables)? If yes, could you provide statistics or examples?

3. How are symbolic extensions implemented?

---

> ### Author Response · Authors · 2025-11-21
> **Weakness 1 & Question 1**
>
> We thank the reviewer for the thoughtful and constructive feedback. We appreciate the time and care taken to read our work in detail and to identify areas where additional clarification and comparison would strengthen the paper. Below we address each concern point-by-point and describe the revisions we have made accordingly.
>
> > **Weakness 1: Missing related work on multi-table QA**: The paper overlooks directly relevant and recent multi-table QA datasets such as
> >
> > - Wu et al., Evaluating LLMs with Multi-Table Multi-Hop Complex Questions. ICLR 2025.
> > - Singh et al., MTabVQA: Evaluating Multi-Tabular Reasoning of Language Models in Visual Space. arXiv preprint arXiv:2506.11684 (2025)
> >
> > Both address similar goals and should be discussed and compared experimentally or at least conceptually. Without this, the novelty claim is weakened.
> >
> > **Question 1:** How does TQA-Bench compare quantitatively to MMQA and MTabVQA in size, complexity, and domain coverage?
> >
>
> We have added both works to the **Appendix A** and explicitly discuss their relation to TQA-Bench.
>
> **MMQA [1].** MMQA evaluates LLMs on multi-table, multi-hop questions built on top of the Spider databases. Conceptually, it is close to our work in that it targets complex questions over multiple relational tables. However, MMQA remains within the Spider schema, whereas TQA-Bench constructs diverse analytical databases from WorldBank, Data.gov, and BIRD, which differ in domain, scale, and schema design. Moreover, TQA-Bench explicitly controls serialized context length (8K to 64K tokens) and table formats via relational sampling and symbolic extensions, enabling systematic studies of long-context and serialization effects; MMQA does not target controlled context-length sweeps.
>
> **MTabVQA [2].** MTabVQA is a visual multi-tabular QA benchmark designed for vision language models: models are given images of tables (often across multiple sheets) and must answer questions in the visual modality. In contrast, TQA-Bench operates purely in the text modality: we serialize relational tables into Markdown/CSV/JSON/HTML and feed these token sequences directly to LLMs. As such, MTabVQA and TQA-Bench explore orthogonal dimensions of multi-table reasoning---visual versus text-only interfaces. A quantitative comparison would require extending TQA-Bench with a VLM-based evaluation pipeline, which we view as interesting future work rather than the focus of the present paper.
>
> We also summarizes MMQA, MTabVQA, and TQA-Bench in terms of size, complexity, and domain coverage, based on statistics reported in their papers.
>
> | Benchmark | Size | Complexity | Domain Coverage |
> | --- | --- | --- | --- |
> | **MMQA**  | Multi-task benchmark over 3,312 tables from Wikipedia with 3313 questions; includes multi-table QA, retrieval, and text-to-SQL tasks. | Multi-table, multi-hop reasoning over 2–3 tables; have 4 question categories (Numerical, List, Count, Select). | Broad Wikipedia domains; all text-based relational tables. |
> | **MTabVQA**  | 8499 tables with 3,745 QA pairs over multi-tabular **images** from real spreadsheets and documents. | Multi-hop **visual** reason­ing across multiple table images; requires table localization, OCR, and cross-tab aggregation. | Real spreadsheet/document domains (finance, demographics, education, health); all table data presented as images. |
> | **TQA-Bench**  | 10 real-world relational databases; **14,000 QA per context scale** (56,000 in total). | Single- and multi-table analytical QA; avg. 1.2–1.8 tables per query; covers lookup, top-k, count, sum, avg, composite comparison, correlation; targets long-context serialized reasoning. | WorldBank/Data.gov/BIRD domains (airlines, food safety, water quality, biodiversity, restaurants, universities); text-only serialized tables. |
>
> **Reference:**
>
> [1] Wu, Jian, et al. "MMQA: Evaluating LLMs with multi-table multi-hop complex questions." *The Thirteenth International Conference on Learning Representations*. 2025.
>
> [2] Singh, Anshul, Chris Biemann, and Jan Strich. "MTabVQA: Evaluating Multi-Tabular Reasoning of Language Models in Visual Space." *arXiv preprint arXiv:2506.11684* (2025).

---

> ### Author Response · Authors · 2025-11-21
> **Weakness 2 & Question 2**
>
> > **Weakness 2: Dataset scope and representativeness**: The included databases (especially those from BIRD) are highly structured and relational, which may not reflect the semi-structured nature of many real-world tables commonly used in reasoning tasks. This limits the generality of conclusions and may bias the benchmark toward text-to-SQL-style reasoning. A clearer discussion contrasting structured (relational) vs. semi-structured table QA would be valuable.
> >
> > **Question 2:** Are any of the databases semi-structured (e.g., with missing relations, nested tables)? If yes, could you provide statistics or examples?
> >
>
> Our current goal is to systematically evaluate LLMs on multi-table QA over *relational* databases, with tight control over context length and serialization format. As described in **Section 2**, all databases in TQA-Bench (WorldBank, Data.gov, BIRD) are fully relational and come with explicit foreign-key graphs, which are crucial for our sampling procedure and symbolic extensions. Thus, in the sense of the question, none of the databases in TQA-Bench are semi-structured.
>
> We agree that many real-world table reasoning tasks involve semi-structured data, and that this is an important but quite broad design space. Designing a unified sampling and serialization pipeline for heterogeneous semi-structured sources is orthogonal to the present work, and we view it as an interesting extension for future benchmarks. In the current paper, we only touch this axis indirectly via our format comparison in **Section 4.2**: JSON and HTML provide more verbose, tree-like serialized structures than Markdown/CSV, and we observe that JSON is consistently the weakest format across models and context lengths (Tables 3–4). This suggests that semi-structured *encodings* of otherwise relational data can already be challenging for current LLMs.
>
> In the revised version, we make this discussion explicit in **Section 4.2**, framing our JSON/HTML results as preliminary evidence that more semi-structured encodings can be harder for LLMs, while clearly stating that TQA-Bench does not yet cover genuinely semi-structured table *sources* (e.g., web tables or document-style JSON stores).

---

> ### Author Response · Authors · 2025-11-21
> **Weakness 3**
>
> > **Weakness 3: Lack of dataset analysis**: The paper spends much of its space on model results (Tables 3–5) but omits essential dataset diagnostics, such as the number of databases, query types per complexity class, or distribution of join depth. Such analysis is necessary to assess the diversity and difficulty of TQA-Bench.
> >
>
> We agree that the current draft does not present enough dataset analysis. In the current draft, we mainly focus on model results and do not expose enough diagnostics of the benchmark itself.
>
> **Database and question composition.** TQA-Bench is built from 10 relational databases sampled from WorldBank, Data.gov and BIRD. For each database, we define 14 question templates covering 7 subcategories (entity lookup, top selection, count, sum, average, composite comparison, correlation), with 2 templates per subcategory. Each template is instantiated into multiple symbolic question instances for each sampled database instance, yielding balanced question counts across all 7 subcategories in every experiment.
>
> **How many tables each question touches.** We agree that it is important to show how often questions require multi-table reasoning. For each database, we therefore computed the average number of distinct tables referenced per question. As in the following results, the averages range from roughly 1.2 to 1.8 tables per question, indicating a mix of single-table and multi-table queries. We have added this column to the database statistics Table 8 in **Appendix B.1**.
>
> | Database Name | Join Depth |
> | --- | --- |
> | airline | 1.79 |
> | food_inspection | 1.64 |
> | movie | 1.21 |
> | music_tracker | 1.5 |
> | restaurant | 1.64 |
> | university | 1.71 |
> | cookbook | 1.43 |
> | food_facility_inspections | 1.64 |
> | water_quality | 1.21 |
> | global_biodiversity | 1.71 |
>
> These additions will make the diversity and difficulty of TQA-Bench more transparent and clarify that it covers a range of analytical subcategories over both single-table and multi-table queries, rather than being dominated by very shallow lookups.

---

> ### Author Response · Authors · 2025-11-21
> **Weakness 4**
>
> > **Weakness 4**: **Formatting and presentation**: While generally clear, several sections are overly detailed (e.g., exhaustive model tables) at the expense of key conceptual comparisons. The narrative could be improved by focusing more on dataset design choices and benchmark positioning rather than numerical results alone.
> >
>
> It is a helpful suggestion regarding the balance between numerical results and conceptual comparisons. Our intention in including exhaustive model tables (e.g., Table 5 and Figure 3) was to ensure transparency and allow practitioners to inspect per-model performance across subcategories and context lengths. However, we agree that in the current draft these detailed tables may draw attention away from the higher-level conceptual insights, as well as from the core design choices and positioning of TQA-Bench.
>
> In the revised version, we will (i) move part of the long per-model result tables to the appendix and keep a more compact summary in the main text, and (ii) strengthen the narrative around dataset design and benchmark positioning. Concretely, we will expand **Section 2** and the related work section to more clearly articulate the design rationale of our sampling procedure, symbolic extensions, and task taxonomy, and to better contrast TQA-Bench with existing (mostly single-table) TableQA benchmarks. We will also revise **Section 4** so that each experiment’s discussion explicitly highlights the key conceptual comparisons rather than focusing primarily on reporting raw accuracies.
>
> We believe these edits will improve the readability and storytelling of the paper, while still preserving the completeness of the empirical evaluation for readers who are interested in detailed model-wise numbers.

---

> ### Author Response · Authors · 2025-11-21
> **Weakness 5**
>
> > **Weakness 5**: **Unclear size compression explanation**: The token count for Markdown, CSV, and JSON in Table 3 decreases from 8K to smaller values (5.4k, 3.7k, 5.75k tokens), which seems counterintuitive. Clarification or an example of the serialization process would help ensure reproducibility.
> >
>
> Our “8K / 16K / 32K / 64K” labels denote context-length budgets, not exact serialized lengths. For each scale, we first sample databases only in Markdown so that the Markdown serialization of all tables falls into a target range (Table 9 in **Appendix B.2**): e.g., for the “8K” scale we sample instances whose Markdown length lies between 4k and 6k tokens, leaving headroom for the natural-language question, prompt, and potential intermediate content.
>
> Table 3 then reports the empirical average token length after re-serializing the same sampled instances into CSV/JSON/HTML. Because CSV is more compact than Markdown and HTML is more verbose, their average lengths are respectively below and above the Markdown range. Thus the decrease from the nominal “8K” budget to 5.4k/3.7k tokens is expected and reflects our design choice to under-saturate the budget in Markdown and to compare different formats under the same underlying tables.

---

> ### Author Response · Authors · 2025-11-21
> **Question 3**
>
> > **Question 3**: How are symbolic extensions implemented?
> >
>
> The full procedure of the symbolic extensions is described in **Appendix B.4** and implemented in our released generation scripts.
>
> Concretely, symbolic extensions are implemented as a rule-based Python generator with two stages:
>
> - **Symbolic templates and gold-answer computation.** For each database and each subcategory, we manually design a symbolic question template where all database-dependent quantities are placeholders. At generation time, we sample rows from the sampled relational instance, fill these placeholders with real cell values, and compute the gold answer by running the corresponding Python code over the tables (relational joins, filters, aggregation or correlation as specified in Table 1). This yields multiple instantiated questions from a single template, as illustrated for the “airline” database in Figures 1 and 5.
> - **Multiple-choice option construction.** Each instantiated question is converted into an MCQ. For entity-lookup and top-selection tasks, we generate incorrect options by randomly sampling different cells from the same column as the correct answer in the same table. For most numeric tasks (count, sum, average, composite comparison), we create three distractors by multiplying the correct answer by fixed factors 0.25, 2.0, and 3.0 and discarding any duplicates or values that coincide with the gold answer. Finally, we shuffle the correct answer and distractors to form the final option set. As for correlation task, we use different factors depending on the answer to make sure all distractors are between -1 and 1.
>
> Using this pipeline, each template is instantiated on ten sampled database instances per context length, and each template yields ten question instances, resulting in a large and diverse pool of questions (see **Appendix B.4** and Table 11 for statistics).
>
> Although the distractor generation rules are simple, our experiments show that the resulting questions are still challenging for many LLMs, especially in aggregation and complex calculation categories.

---

### Official Review · Reviewer_rikH · 2025-10-29

**Soundness:** 2
**Presentation:** 2
**Contribution:** 2
**Rating:** 4
**Confidence:** 4

**Summary:**

TQA-Bench proposes a multi-table QA benchmark that (1) collects relational databases from WorldBank / Data.gov / BIRD, (2) samples those databases to produce serialized multi-table contexts at multiple lengths (8K → 64K tokens), (3) generates templated questions augmented with “symbolic extensions” and Python answer generation, and (4) evaluates a wide range of LLMs (open + closed source, ~2B–671B params) under different serialization formats and with both direct prompting and Text2SQL approaches. Key reported findings include Markdown being the best serialization, instruct models outperforming chat models, large performance drops as context grows, and Text2SQL being more stable but struggling on complex analytics.

**Strengths:**

Important problem & scale — it addresses an under-explored but high-impact task (multi-table QA across large relational contexts) and explicitly targets large serialized contexts (8K–64K tokens), which is timely as long-context LLMs become common

Diverse, public data sources — the benchmark draws from WorldBank, Data.gov, and BIRD (not Wikipedia), which increases variety and reduces simple contamination from wiki tables. The curated selection and reasoning about referential integrity are sensible.

the paper evaluates many models (28) spanning architectures, sizes, and closed/open source; runs format (Markdown/CSV/JSON/HTML) comparisons; contrasts single- vs multi-table; and compares direct prompting vs Text2SQL. That breadth yields many actionable observations.

**Weaknesses:**

W1: The paper reports many percent accuracies and heatmaps, and it mentions generating “multiple instances to reduce variance,” but I found no paired statistical tests, confidence intervals, or significance testing for model comparisons (e.g., better vs baseline). It’s unclear how stable reported differences (e.g., Markdown vs CSV) are


W2: The authors intentionally excluded Wikipedia but still use WorldBank / Data.gov / BIRD — public sources that may be present in LLM pretraining corpora. The paper claims sampling reduces leak, but gives no contamination analysis (e.g., checking if test templates/queries or answer strings appear in model pretraining corpora or online).


W3: The evaluation uses a single-choice format (A/B/C/D) with prompts that insist on a single letter answer. This punishes partially correct outputs or correct numeric answers that were phrased differently; at the same time, the Text2SQL evaluation requests the chain-of-thought and SQL code. For complex analytical tasks (correlation, averages), multiple formats for correct answers exist.


W4: The paper gives overall accuracy and heatmaps but lacks a deep qualitative error analysis on why top models fail on correlation and composite tasks. There are mentions (e.g., SQL composition difficulties) but not many concrete failure examples analyzed in detail.


W5: Metric clarity: Clarify how average scores are computed across subcategories (micro vs macro averages). Provide a full metric table with counts per subcategory.

**Questions:**

same as weakness

---

> ### Author Response · Authors · 2025-11-21
> **Weakness 1**
>
> We sincerely thank the reviewer for the careful reading of our manuscript and the thoughtful, constructive suggestions. The comments have been extremely valuable in helping us refine both the presentation and positioning of the work. We address each point in detail below and outline the specific revisions incorporated into the updated manuscript
>
> > **Weakness 1**: The paper reports many percent accuracies and heatmaps, and it mentions generating “multiple instances to reduce variance,” but I found no paired statistical tests, confidence intervals, or significance testing for model comparisons (e.g., better vs baseline). It’s unclear how stable reported differences (e.g., Markdown vs CSV) are
> >
>
> We add paired statistical tests and confidence intervals in our original format comparison. In addition to the accuracy numbers in Table 4, we now provide a more systematic analysis of the key trade-off between Markdown and CSV, which are the practically relevant formats in our setting (HTML has severe token overhead, and JSON does not yield better accuracy–length trade-offs).
>
> **1. Paired McNemar tests (Markdown vs. CSV)**
>
> For each model and context length, we run a McNemar test on paired per-question correctness between Markdown and CSV. Each question instance provides a binary outcome (correct / incorrect) under both formats, and the test is conducted with one degree of freedom. The resulting $\chi^2$ statistics and $p$-values are:
>
> | Model | Context | chi2 | p-value |
> | --- | --- | --- | --- |
> | Qwen2.5 | 8k | 19.79 | 8.65e-6 |
> |  | 16k | 42.00 | 9.11e-11 |
> |  | 32k | 13.32 | 2.63e-3 |
> |  | 64k | 0.11 | 0.74 |
> | Qwen2.5-Coder | 8k | 0.019 | 0.89 |
> |  | 16k | 3.54 | 0.060 |
> |  | 32k | 22.44 | 2.17e-6 |
> |  | 64k | 0.11 | 0.74 |
> | Llama3.1 | 8k | 1.14 | 0.29 |
> |  | 16k | 4.72 | 0.030 |
> |  | 32k | 1.31 | 0.25 |
> |  | 64k | 0.0 | 1.0 |
>
> The pattern aligns closely with the raw accuracy gaps in Table 4:
>
> - **Qwen2.5-7B-Instruct:** Markdown significantly outperforms CSV at 8k, 16k, and 32k ($p\ll0.01$).
> - **Qwen2.5-Coder-7B-Instruct:** We see a strong and significant Markdown advantage at 32k.
>
> We can find that, the large observed Markdown–CSV gaps in Table 4 (Qwen2.5 at 8k/16k/32k, Qwen2.5-Coder at 32k) correspond to large $\chi^2$ and very small $p$-values.
>
> **2. Accuracy and 95% confidence intervals**
>
> To quantify the uncertainty of the reported accuracies, we also compute 95% binomial confidence intervals for Markdown and CSV at each context scale. The results are:
>
> | Model,Format | Format | 8k | 16k | 32k | 64k |
> | --- | --- | --- | --- | --- | --- |
> | Qwen2.5 | MD | 49.86(47.97-51.75) | 50.86(48.97-52.75) | 41.57(39.71-43.43) | 33.43(31.65-35.21) |
> |  | CSV | 40.57(38.71-42.43) | 36.86(35.04-38.68) | 33.71(31.92-35.50) | 32.57(30.80-34.34) |
> | Qwen2.5-Coder | MD | 48.0(46.11-49.89) | 47.14(45.25-49.03) | 44.86(42.98-46.74) | 36.86(35.04-38.68) |
> |  | CSV | 47.57(45.68-49.46) | 43.0(41.13-44.87) | 34.43(32.63-36.23) | 36.0(34.19-37.81) |
> | Llama3.1 | MD | 46.57(44.68-48.46) | 43.0(41.13-44.87) | 35.43(33.62-37.24) | 35.86(34.05-37.67) |
> |  | CSV | 44.29(42.41-46.17) | 38.71(36.87-40.55) | 33.0(31.22-34.78) | 35.71(33.90-37.52) |
>
> The intervals are fairly tight (typically within ±1–2.5 percentage points). We observe that:
>
> - In settings where McNemar’s test finds a **significant** Markdown–CSV difference (Qwen2.5 at 8k/16k/32k; Qwen2.5-Coder at 32k), the confidence intervals for Markdown and CSV **do not overlap**, reinforcing that the Markdown advantage is statistically robust.
> - In settings where McNemar’s test is **non-significant** (e.g., Qwen2.5 at 64k), the two intervals overlap substantially, indicating that Markdown and CSV are statistically indistinguishable there.
>
> **3. Takeaways for the benchmark design**
>
> These additional diagnostics address the reviewer’s concern about the stability of our reported differences:
>
> - They show that Markdown’s advantage over CSV is **statistically robust** in several core model–context configurations, exactly where the raw accuracy gaps are substantial.
> - At the same time, they clarify that Markdown does **not** uniformly dominate CSV across all models and scales; CSV remains competitive (and sometimes indistinguishable) when the gaps are small.
>
> Considering both accuracy and context length, we therefore continue to use Markdown as the default serialization format for TQA-Bench, while explicitly acknowledging that CSV is a viable alternative in conditions where the accuracy difference is minor.
>
> For completeness, the above McNemar statistics and confidence intervals have also been integrated into the updated manuscript as a dedicated appendix section (“Statistical robustness of format comparison”, **Appendix H**), so that readers can cross-check the exact numbers alongside Table 4.

---

> ### Author Response · Authors · 2025-11-21
> **Weakness 2**
>
> > **Weakness 2**: The authors intentionally excluded Wikipedia but still use WorldBank / Data.gov / BIRD — public sources that may be present in LLM pretraining corpora. The paper claims sampling reduces leak, but gives no contamination analysis (e.g., checking if test templates/queries or answer strings appear in model pretraining corpora or online).
> >
>
> We admit that our current wording overstates what we can guarantee. In particular, the sentence in **Section 1** that we “effectively sample a brand-new benchmark to avoid data leaking” is too strong. We revised the text to state that our design *mitigates* contamination risk and makes it easy to *regenerate new evaluation splits* if contamination is suspected, rather than claiming that it fully avoids leakage.
>
> Our decision to exclude Wikipedia tables is motivated by the fact that Wikipedia is a well-known core component of many LLM pre-training corpora and is also the dominant source for prior table-QA benchmarks (Table 7, **Appendix A**). Using non-Wikipedia sources (WorldBank, Data.gov, BIRD) does **not** guarantee that the underlying tables never appear in pre-training data. The main benefit of these sources is that they (i) reduce reliance on the most heavily reused benchmark tables from Wikipedia, and (ii) provide large, relational, long-context schemas that better match our multi-table evaluation goals.
>
> Regarding the claim that sampling and symbolic extensions “reduce leak,” our intention is not to assert that the current test split is provably uncontaminated, but to highlight two robustness properties of the benchmark: (1) questions and answers are generated by sampling relational subgraphs and recomputing derived quantities (sums, averages, correlations, etc.), so exact question–answer pairs are unlikely to coincide with static facts memorized from raw tables; and (2) because the whole pipeline is programmable, one can re-sample database instances and regenerate question instances with new random seeds if evidence of contamination emerges. We explicitly rephrase this in **Section 1** and **Appendix A**, and add a short limitation sentence stating that we do **not** perform a full pre-training-corpus contamination analysis (which is infeasible for proprietary LLMs), and that our design should be viewed as *mitigating* contamination and enabling easy regeneration of fresh evaluation sets, rather than eliminating it.

---

> ### Author Response · Authors · 2025-11-21
> **Weakness 3**
>
> > **Weakness 3**: The evaluation uses a single-choice format (A/B/C/D) with prompts that insist on a single letter answer. This punishes partially correct outputs or correct numeric answers that were phrased differently; at the same time, the Text2SQL evaluation requests the chain-of-thought and SQL code. For complex analytical tasks (correlation, averages), multiple formats for correct answers exist.
> >
>
> Reviewer concerns that our single-choice format with “output a single letter” prompts might penalize otherwise correct answers. To quantify how often this actually happens, we conducted a manual error analysis focused on a strong model in our study. We randomly sampled **50 mispredictions** made by **GPT-4o** in the **8K** scale and inspected both the intermediate reasoning and the final outputs. The outcomes are summarized below:
>
> | Pattern | Count (out of 50) | Description |
> | --- | --- | --- |
> | Wrong letter consistent with wrong reasoning | 46 | Model outputs a well-formed A/B/C/D choice that matches its (incorrect) reasoning. |
> | Non-letter answers contradicting task setup (e.g., “None”) | 4 | Model confidently claims that no option is correct, although exactly one is by design. |
> | Correct numeric answer but wrong/absent letter | 0 | Model arrives at the ground-truth numeric value but loses credit only due to format. |
>
> In **all 50** examined cases, the model’s reasoning or final numerical conclusion was incorrect relative to the ground-truth answer. We did **not** observe any instance where a correct numeric answer was scored as wrong solely because it was not mapped to the corresponding letter. This suggests that, at least for a strong model like GPT-4o at 8K, the dominant source of errors is the underlying analytical reasoning rather than the single-letter answer format.
>
> We agree that more fine-grained grading schemes (e.g., parsing free-form numeric outputs or giving partial credit for nearly-correct expressions) are an interesting direction, especially for complex analytic queries. However, implementing and validating such task-specific graders at the scale of **tens of thousands of questions** is non-trivial and orthogonal to the main goal of TQA-Bench. We therefore adopt a **single-letter, exact-match** protocol that enables scalable, objective, and unambiguous automatic evaluation, in line with widely used multiple-choice LLM benchmarks such as MMLU and MathQA.
>
> For the Text2SQL setting, although the prompt allows chain-of-thought and SQL code, our metric is likewise a **0/1 exact-match on the executed result**: we run the generated SQL and compare the returned value to the ground-truth answer, without awarding partial credit for partially correct reasoning or almost-correct queries. The richer output format there is used only to help models produce executable SQL, not to relax the evaluation criterion.
>
> We have added the above error analysis and clarification to **Appendix G.1** in the revised manuscript so that readers can verify the details.

---

> ### Author Response · Authors · 2025-11-21
> **Weakness 4**
>
> > **Weakness 4**: The paper gives overall accuracy and heatmaps but lacks a deep qualitative error analysis on why top models fail on correlation and composite tasks. There are mentions (e.g., SQL composition difficulties) but not many concrete failure examples analyzed in detail.
> >
>
> We agree that it is important to provide a deeper qualitative analysis of why top models fail on correlation and composite analytical tasks. To address this, we performed a manual error analysis focused on the **Text2SQL** setting and the hardest task types. Specifically, we sampled **50 mispredicted queries** produced by **DeepSeek-R1** at the **8K context length**, restricted to **complex calculation questions**. From these 50 cases, we identified three recurring error patterns:
>
> - **(1) Non-executable or non-SQL output (≈26–52%)**
>
>     In many correlation (COR) questions, the model outputs a mathematically meaningful *formula* instead of an executable SQL query. For example, it may return an expression of the form `correlation = (n * sum_xy - sum_x * sum_y) / SQRT((n * sum_x2 - sum_x^2) * (n * sum_y2 - sum\_y^2))` as its “final SQL query”. This confirms that the model often understands the *statistical* operation conceptually, but fails to translate it into valid SQL that can be run by the database engine.
>
> - **(2) Wrong solution step or task formulation (≈16–32%)**
>
>     Here, the model produces syntactically valid SQL, but the query implements the *wrong* operation for the question. For instance, for the question “How many budgets is *Remember the Titans* higher than *X-Men Origins: Wolverine*?”, one generated query simply checks whether one budget is larger than the other and returns a binary indicator (1 or 0), instead of computing the numeric **difference** between the two budgets. In other words, the model locates the right tables and columns, but it chooses the wrong computational step (comparison vs. difference).
>
> - **(3) Correct high-level plan but mis-specified expressions (≈8–16%)**
>
>     In these cases the model outlines a reasonable multi-step SQL strategy, but small expression-level choices cause a mismatch with the benchmark’s intended semantics. For example, for a question asking for the average total fly time defined as `ARR_TIME − DEP_TIME`, the model tries to interpret `ARR_TIME` and `DEP_TIME` in HHMM format, converts them to minutes via `t/100*60 + t%100`, and adds a 1440-minute wrap-around for overnight flights. This “overly careful” time-handling logic is plausible in isolation, but it **does not match** the task definition used in our benchmark (direct difference in the stored integers), and thus leads to an incorrect answer.
>
>
> Overall, these qualitative patterns support our main claim that the bottleneck for top Text2SQL models on correlation and composite tasks lies in **composing the correct sequence of SQL operations and matching the benchmark’s problem formulation**, rather than merely recalling individual operators or column names. Models typically identify the relevant tables and fields, but struggle to (i) turn statistical formulas into executable SQL, (ii) select the correct computational step (e.g., difference vs. comparison), and (iii) align detailed arithmetic/time handling with the task definition.
>
> We have added this qualitative analysis, together with the concrete examples summarized above, to **Appendix G.2** in the revised manuscript so that readers can inspect the exact queries and outputs in more detail.

---

> ### Author Response · Authors · 2025-11-21
> **Weaknesss 5**
>
> > **Weakness 5**: Metric clarity: Clarify how average scores are computed across subcategories (micro vs macro averages). Provide a full metric table with counts per subcategory.
> >
>
> We admit that our metric definition was not sufficiently explicit. We already update the definition in Table 4.
>
> **How averages are computed.** All accuracies are computed at the question level. For each model and condition, we first compute accuracy within each of the 7 task subcategories. The reported “Average” column is the unweighted mean across subcategories. Because our benchmark design fixes the number of questions per subcategory, this macro-average is numerically equal to the corresponding micro-average over all questions.
>
> **Why subcategory counts are balanced.** As described in **Appendix B.3–B.4**, for each database we manually design two template questions per subcategory, and each template is instantiated into 10 symbolic question instances for each of the 10 sampled database instances. This yields the same number of questions for every subcategory in every experiment (14 templates per database × 10 question instances × 10 databases × 10 database instances = 14,000 questions per context or format condition). Hence no subcategory dominates the overall score, which was our motivation for combining sampling with symbolic extension.

---

### Official Review · Reviewer_6s7s · 2025-11-01

**Soundness:** 2
**Presentation:** 2
**Contribution:** 2
**Rating:** 4
**Confidence:** 4

**Summary:**

This paper presents a new benchmark for evaluating large language models over TableQA tasks, focusing on multi-table question answering settings. Detailed dataset construction procedures are presented, where a sampling strategy is proposed to preserve the foreign key dependencies between tables in a database, and a template-based question generation strategy is proposed to generate questions together with ground-truth answers. Experimental results with 28 large language models are presented, with a few interesting findings such as markdown being the most effective input format when feeding tables into large language models and that direct table prompting consistently outperforms Text2SQL-based solutions in accuracy.

**Strengths:**

1. This paper presents a new multi-table TableQA dataset of varying scales which could be useful for follow-up studies.

2. A large number (28) of large language models are tested over the proposed benchmark dataset.

3. There are some interesting findings in the reported results, e.g., markdown is the most effective input format for feeding tables into large language models; and direct table prompting consistently outperforms Text2SQL-based solutions in accuracy.

4. Source code and datasets are included in the submission.

**Weaknesses:**

1. Novelty:

- The proposed new TableQA benchmark is somewhat incremental. While its scale is larger in terms of the table/database sizes (which is relatively easy to achieve), there is limited variety in the questions (14 template questions) which does not make a particularly interesting dataset.

- Some of the findings are not particularly interesting, e.g., LLMs perform better when the question related tables are precisely identified and given to the LLMs (instead of all tables in a database), although the paper may still serve as empirical evidence for future studies.

2. Technical details:

- Table 7 suggests that the constructed dataset has 8K to 64K tokens per **table** on average, while Table 10 suggests that these are the average number of tokens per **database**. Please clarify which is intended.

- It is not quite sure what is the intended calculation target of the correlation queries.

- In Table 11, it is not quite sure why there are 14,000 question instances under each context length setup. There are 100 database instances (= 10 different databases x 10 sampled instances per database), and there are 14 questions per database (= 2 template questions per question subcategories x 7 subcategories). It seems 100 x 14 = 1,400 question instances to me.

- While there are 28 LLMs involved in the experiments in total. Each set of the experiments uses a different subset of LLMs. The rationale behind the choice of LLMs for each experiment is unclear. The separation of experimental setup from the results of each experiment makes the paper unnecessarily difficult to follow.

3. Minor presentation issues:

- "The rise of large language models (LLMs) Jin et al. (2022)" => "The rise of large language models (LLMs) (Jin et al. 2022)"

- "Average Columns" => "Average #Columns", "sampled row number" => "sampled number of rows"

- "greater or equal than" => "greater than or equal to"

- "department delay" => "departure delay"

- What does it mean by "average total delay"?

- Published version of Tablebench should be cited instead of the arxiv version.

**Questions:**

See Weaknesses 1 & 2.

---

> ### Author Response · Authors · 2025-11-21
> **Weakness 1**
>
> We thank the reviewer for the detailed and insightful comments. We greatly appreciate the effort in highlighting both strengths and areas where further clarification would improve the clarity and completeness of the paper. We have revised the manuscript accordingly and respond to each point below.
>
> > **Weakness 1**: Novelty
> **a).** The proposed new TableQA benchmark is somewhat incremental. While its scale is larger in terms of the table/database sizes (which is relatively easy to achieve), there is limited variety in the questions (14 template questions) which does not make a particularly interesting dataset.
> >
>
> Regarding the concern that the benchmark has “only 14 template questions,” we acknowledge that our writing may not have conveyed this aspect clearly, which could have caused misunderstanding. For each database, we manually design two templates per subcategory across seven subcategories (entity lookup, top selection, count, sum, average, composite comparison, and correlation), yielding 14 templates *per database* and 140 templates in total across the ten heterogeneous real-world databases. These templates are domain-specific (airline, restaurant, water quality, global biodiversity, etc.) and are paired with Python implementations. For each database and context length, we sample ten database instances, and via symbolic extension we instantiate each template into ten concrete questions (e.g., varying entities and thresholds), resulting in 100 database instances and 14,000 question instances per context length, or 56,000 questions overall. This design focuses on a controlled yet broad taxonomy of relational operations over multi-table schemas, rather than on an unbounded number of surface patterns, and enables systematic cross-database and cross-context comparisons. In the revised version, we will make the number of templates and instances explicit in the main text and add brief statistics on the distribution of question types to better convey this diversity.
>
> > **b).** Some of the findings are not particularly interesting, e.g., LLMs perform better when the question related tables are precisely identified and given to the LLMs (instead of all tables in a database), although the paper may still serve as empirical evidence for future studies.
> >
>
> We agree with the reviewers that some findings (e.g., models perform better when given only the question-relevant tables) are intuitive. These experiments are not intended as conceptual contributions; they serve as ablations/sanity checks to validate that our multi-table setup is indeed sensitive to schema linking and table selection, and to quantify how large this effect is across model families. Notably, the magnitude of this improvement—ranging from 6% to 20% across models—was larger than we initially expected, suggesting that schema selection remains a non-trivial bottleneck even for strong LLMs. Thus, although intuitive, these results provide useful empirical evidence for future studies on multi-table reasoning, particularly regarding how LLMs handle schema isolation versus full-schema contexts.
>
> Importantly, even in the “relevant-table” setting, performance on complex analytical tasks such as composite comparison and correlation remains far below that on simple lookup/aggregation, indicating that the main bottleneck lies in multi-step reasoning and correct grounding rather than table retrieval alone. We will clarify this positioning in the paper by explicitly labeling these studies as ablation/sanity-check experiments and stressing that the main contributions lie in (i) constructing long-context, multi-table serialized databases (up to 64K tokens per database instance), (ii) defining a RA-grounded task taxonomy including correlation, and (iii) systematically evaluating 28 LLMs under this setting.

---

> ### Author Response · Authors · 2025-11-21
> **Weakness 2**
>
> > **Weakness 2**: Technical details
> **a).** Table 7 suggests that the constructed dataset has 8K to 64K tokens per **table** on average, while Table 10 suggests that these are the average number of tokens per **database**. Please clarify which is intended.
> >
>
> In Table 7, for prior work we report the average token count *per table*, whereas for TQA-Bench we summarize the *per-database* serialized context length scales (“8K–64K”), since our evaluation feeds the entire multi-table database (with table names and schema) to the model. The detailed per-database statistics already appear in Table 10. We already revised the text and header around Table 7 to explicitly state that our row refers to tokens per serialized database instance, to avoid confusion.
>
> > **b).** It is not quite sure what is the intended calculation target of the correlation queries.
> >
>
> For the correlation (COR) subcategory, our intention is to compute the standard Pearson correlation coefficient between two numeric columns after applying the specified join and filter; **Appendix B.3** in the revised manuscript now includes the explicit formula.
>
> > **c).** In Table 11, it is not quite sure why there are 14,000 question instances under each context length setup. There are 100 database instances (= 10 different databases x 10 sampled instances per database), and there are 14 questions per database (= 2 template questions per question subcategories x 7 subcategories). It seems 100 x 14 = 1,400 question instances to me.
> >
>
> The numbers in Table 11 follow directly from the construction above: for each context length, we have 10 databases × 10 sampled instances × 14 templates × 10 symbolic instantiations = 14,000 question instances.
>
> > **d).** While there are 28 LLMs involved in the experiments in total. Each set of the experiments uses a different subset of LLMs. The rationale behind the choice of LLMs for each experiment is unclear. The separation of experimental setup from the results of each experiment makes the paper unnecessarily difficult to follow.
> >
>
> Regarding the use of different subsets of 28 LLMs across experiments, our evaluation is intentionally staged under a shared compute budget: Experiments 1–2 are design/calibration studies (single vs multi-table; serialization formats), where a small set of representative models (chat vs instruct vs coder/reasoning) suffices to answer the methodological questions. Experiments 3–5 then fix the design choices and perform broader comparisons across model families and context lengths. In the revised manuscript we add short “LLM selection rationale” paragraphs in **Appendix D.1–D.5** to make this staging explicit and to document which models are used where and why.

---

> ### Author Response · Authors · 2025-11-21
> **Weakness 3**
>
> > **Weakness 3**: Minor presentation issues:
> >
> > - "The rise of large language models (LLMs) Jin et al. (2022)" => "The rise of large language models (LLMs) (Jin et al. 2022)"
> > - "Average Columns" => "Average #Columns", "sampled row number" => "sampled number of rows"
> > - "greater or equal than" => "greater than or equal to"
> > - "department delay" => "departure delay"
> > - What does it mean by "average total delay"?
> > - Published version of Tablebench should be cited instead of the arxiv version.
>
> In the revised version, we have addressed all noted presentation issues:
>
> **a).** corrected the citation style (“The rise of large language models (LLMs) (Jin et al. 2022)”)
>
> **b).** updated table headers from “Average Columns” to “Average #Columns”, change from “sampled row number” to “sampled number of rows”
>
> **c).** fixed the phrasing “greater or equal than” to “greater than or equal to”
>
> **d).** corrected “department delay” to “departure delay”.
>
> **e).** updated the citation of TableBench to its published version instead of the arXiv preprint.
>
> Regarding “average total delay” in Figure 1, we agree that the phrase “total delay” is ambiguous in everyday language: depending on context, it could refer to the sum of different delay components, a net difference, or some other aggregate. This kind of semantic vagueness is precisely what naturally arises in real user queries.
>
> In our benchmark, however, we do not leave this quantity underspecified. In the Figure 1 example, the natural-language phrase is immediately followed by a parenthesized formula that fixes its semantics, i.e., “average total delay (DEP_DELAY − ARR_DELAY)”. Concretely, we first compute `e=DEP_DELAY−ARR_DELAY` for each flight satisfying the filters, and “average total delay” denotes the arithmetic mean of this value `e` over those flights.
>
> Thus, while the English label “total delay” is linguistically ambiguous in isolation, the attached formula in parentheses specifies exactly what is being computed in our benchmark, and this formally defined quantity is what we use for answer generation and evaluation.

---

> > ### Comment · Reviewer_6s7s · 2025-11-26
> >
> > Thank you for the detailed response. My questions regarding technical details and presentation are addressed. I raised my Soundness and Presentation scores to 3. On novelty, while I take the authors' argument, I'm not fully convinced, acknowledging that we may not always agree upon the perceived novelty of a piece of work. My overall rating hence stays unchanged.

---

> > > ### Author Response · Authors · 2025-12-02
> > >
> > > We sincerely thank the reviewer for the constructive feedback and for raising the Soundness and Presentation scores.
> > >
> > > We understand that opinions on novelty can differ and we appreciate the reviewer’s perspective, which will help us further improve the presentation of our contributions.

---

### Official Review · Reviewer_aAd1 · 2025-11-02

**Soundness:** 3
**Presentation:** 3
**Contribution:** 2
**Rating:** 4
**Confidence:** 4

**Summary:**

The paper introduces TQA-Bench, a new benchmark for multi-table question answering. The benchmark is constructed from real-world databases (BIRD, Data.gov, and WorldBank) and features serialized table data ranging from 8K to 64K tokens, specifically targeting long-context capabilities. A key feature is the use of "symbolic extensions" to programmatically generate a diverse set of questions from templates. The authors present a comprehensive evaluation of 28 LLMs on this benchmark, analyzing performance across different model types, serialization formats, and prompting methods.

**Strengths:**

+ A primary strength is the extensive benchmarking of 28 different models, encompassing a wide variety of open-source instruct, chat, and reasoning-focused LLMs (e.g., Qwen, Llama, DeepSeek).

+ The emphasis on long-context performance (8K-64K tokens) is highly relevant. It addresses a notable gap in many existing table QA benchmarks, which often feature much smaller data volumes. This provides valuable insights into how model performance scales or degrades, with increasing context length.

+ The paper provides a thorough analysis of model performance, examining different serialization formats, and various prompting methods (e.g., direct QA vs. text-to-SQL).

**Weaknesses:**

While the benchmark is well-constructed, **a major concern is its long-term utility for the community.** The benchmark may already be approaching saturation or being "readily solved" by current models. The paper acknowledges existing multi-table benchmarks, positioning long-context as the key differentiator. However, the performance of the best-evaluated model (DeepSeek-R1) is already very high: 94.38% at 8K, 89.56% at 32K, and 81.50% at 64K. These scores suggest the benchmark may be too easy for top-tier models. This concern is amplified by the absence of current frontier models (e.g., GPT 5, Gemini 2.5 Pro, Claude 4.1), making it difficult to gauge the benchmark's true "headroom" or difficulty.

The paper identifies the Correlation (COR) task as particularly difficult. **However, it is questionable whether direct prompting or text-to-SQL are the most practical or powerful approaches for such tasks**. A more pragmatic solution, which the benchmark overlooks, would be to use coding agent (an LLM generated and executed code iteratively given observations from the environment). It would be valuable to evaluate models within an agentic framework (e.g., open-source SWE-agent, Qwen-Code, or close but strong Claude Code). Such an agent could write and execute Python code, a trivial way to solve correlation problems, based on feedback from a code interpreter. It might reflect a more realistic and capable problem-solving paradigm.

**Questions:**

Could the authors elaborate more on the benchmark's positioning?

+ How does TQA-Bench's focus on long-context serialized data differ from Spider 2.0, a industry-level complex text-to-SQL benchmark which also target complex database schemas and multi-step reasoning.

+ The distinction between this multi-table QA setup and a more general task, where a coding agent interacts with several local .csv or .db files using Python, is also a bit unclear to me. The latter seems to be a superset of the task proposed in this benchmark.

---

> ### Author Response · Authors · 2025-11-21
> **Weakness 1**
>
> We appreciate the reviewer’s concern regarding the long-term utility of TQA-Bench and the possibility that the benchmark may be approaching saturation, as well as the suggestion to include evaluations of coding-agent systems.
>
> > **Weakness 1**: While the benchmark is well-constructed, **a major concern is its long-term utility for the community.** The benchmark may already be approaching saturation or being "readily solved" by current models. The paper acknowledges existing multi-table benchmarks, positioning long-context as the key differentiator. However, the performance of the best-evaluated model (DeepSeek-R1) is already very high: 94.38% at 8K, 89.56% at 32K, and 81.50% at 64K. These scores suggest the benchmark may be too easy for top-tier models. This concern is amplified by the absence of current frontier models (e.g., GPT 5, Gemini 2.5 Pro, Claude 4.1), making it difficult to gauge the benchmark's true "headroom" or difficulty.
> >
>
> As shown in Table 5, the best-performing model (DeepSeek-R1) indeed achieves very high overall accuracy when averaged across all task types, including relatively easy categories such as Lookup or Aggregation. However, on our most challenging task—**COR** under the **64K context-length** setting—DeepSeek-R1 reaches only **58.33%**, indicating that substantial headroom remains on difficult analytical workloads.
>
> To further assess the benchmark’s difficulty, we additionally evaluate a current frontier model—GPT-5.1 at 64k scale:
>
> | EL | TS | CNT | SUM | AVG | CC | COR | Average |
> | --- | --- | --- | --- | --- | --- | --- | --- |
> | 94.85 | 93.81 | 58.16 | 57.29 | 72.63 | 90.82 | 53.54 | 74.41 |
>
> We observe that **GPT-5.1 obtains only 53.54% on the 64K COR task**, a result comparable to earlier models such as GPT-o3-mini. In other words, despite approximately ten months of rapid model development since DeepSeek-R1’s release (Jan 2025), **frontier LLMs still struggle significantly** on the long-context COR setting.
>
> This lack of improvement suggests that (i) the 64K analytical tasks remain far from being solved, and (ii) current frontier models still exhibit clear weaknesses in long-context multi-table reasoning. Therefore, TQA-Bench is **not saturated** and retains **meaningful long-term utility for the community** as a target for evaluating future progress.
>
> This fact is important and was not sufficiently emphasized in our original writing; we explicitly highlight this point in the **Appendix F**.

---

> ### Author Response · Authors · 2025-11-21
> **Weakness 2**
>
> > **Weakness 2**: The paper identifies the Correlation (COR) task as particularly difficult. **However, it is questionable whether direct prompting or text-to-SQL are the most practical or powerful approaches for such tasks**. A more pragmatic solution, which the benchmark overlooks, would be to use coding agent (an LLM generated and executed code iteratively given observations from the environment). It would be valuable to evaluate models within an agentic framework (e.g., open-source SWE-agent, Qwen-Code, or close but strong Claude Code). Such an agent could write and execute Python code, a trivial way to solve correlation problems, based on feedback from a code interpreter. It might reflect a more realistic and capable problem-solving paradigm.
> >
>
> We thank the reviewer for this valuable suggestion. We acknowledge that the original draft did not include a discussion of more powerful agentic paradigms, and we will add this component as a dedicated discussion in **Appendix F** (Section “Frontier Model and Agent Testing”).
>
> We further clarify our position on evaluating models within an agentic framework. While we fully agree with the reviewer that testing models in end-to-end agent systems is valuable, current agent research offers *no standardized or agreed-upon agent workflow*. As highlighted by FDABench, existing studies propose many heterogeneous agent designs and even the four representative patterns summarized by FDABench fail to converge to a universal “standard agent” [1]. This makes a direct comparison of full agent pipelines difficult, as differences in workflow, tool routing, or planning logic can dominate the results and obscure the effect of the underlying model.
>
> To ensure a fair and interpretable evaluation, we therefore focus on a more fundamental experimental question:
>
> > Within an agent system, which implementation of the table-analysis component (the table agent) is most effective?
> >
>
> This choice is motivated by the nature of multi-agent systems, where complex tasks are decomposed into specialized sub-agents; improving the performance of such a specialized table-analysis component directly enhances the overall agent’s capability. Improving this component directly influences the performance of the entire agent system, while avoiding confounding factors caused by heterogeneous agent workflows.
>
> For this reason, we isolate the table agent and directly compare two commonly used and practically important implementations:
>
> - **(i) direct prompting**, and
> - **(ii) code interpreter (Python Engine)**.
>
> This decomposition allows us to study the reviewer’s suggestion—agentic reasoning using executable code—in a controlled, principled, and comparable manner, without relying on any specific agent workflow design.
>
> **COR task: direct prompting vs. code interpreter**
>
> For the COR task, we compare GPT-5.1 (64K) under direct prompting versus code interpreter, and observe that replacing direct prompting with code interpreter yields substantial improvements.
>
> | direct prompting | code interpreter |
> | --- | --- |
> | 53.54 | 60.53 |
>
> **Summary of our findings**
>
> - **LLM + code interpreter**:
>
>     Performs strongly across context lengths and excels at analytical tasks such as correlation.
>
> - **Direct prompting**:
>
>     Strong for short contexts and simple reasoning, with minimal engineering overhead.
>
>
> Overall, as agentic AI paradigms continue to evolve (e.g., iterative reasoning, multi-step planning, tool use), our experiments could provide empirical evidence for how future agent systems may choose or design their table-analysis component.
>
> **Reference:**
>
> [1] Wang, Ziting, et al. "FDABench: A Benchmark for Data Agents on Analytical Queries over Heterogeneous Data." *arXiv preprint arXiv:2509.02473* (2025).

---

> ### Author Response · Authors · 2025-11-21
> **Question 1**
>
> > **Question 1**: How does TQA-Bench's focus on long-context serialized data differ from Spider 2.0, a industry-level complex text-to-SQL benchmark which also target complex database schemas and multi-step reasoning.
> >
>
> TQA-Bench fills a specific gap in the current benchmark landscape by providing a controlled evaluation of long-context multi-table reasoning without external tools, which is not the focus of existing table QA or Text2SQL datasets.
>
> Spider 2.0 is primarily an enterprise Text2SQL benchmark designed for agentic workflows: models interact with large production-style databases, possibly multiple schemas and SQL dialects, and must plan and execute multi-step query pipelines through a database engine. In this setting, tables are not fully serialized into the prompt; the focus is on schema understanding, SQL planning, and tool use in a realistic environment, rather than on reasoning over long serialized table contexts.
>
> In contrast, TQA-Bench is a multi-table QA benchmark that deliberately removes the external DB / tool dimension and instead stresses long-context reasoning over serialized relational data. For each instance we materialize a snapshot of several related tables into a single prompt (8K–64K tokens) using a chosen serialization format and ask the model to directly output the final answer, with Text2SQL only used as an optional variant in Experiment 5. This design allows us to systematically study (i) single- vs. multi-table difficulty, (ii) the impact of serialization format, and (iii) robustness to context length under fully observable multi-table inputs, which are not the main targets of Spider 2.0.
>
> In this sense, the two benchmarks are complementary: Spider 2.0 evaluates agentic Text2SQL workflows over industrial databases, whereas TQA-Bench isolates end-to-end long-context multi-table reasoning.

---

> ### Author Response · Authors · 2025-11-21
> **Question 2**
>
> > **Question 2**: The distinction between this multi-table QA setup and a more general task, where a coding agent interacts with several local .csv or .db files using Python, is also a bit unclear to me. The latter seems to be a superset of the task proposed in this benchmark.
> >
>
> Regarding the comparison to a general “coding agent over local .csv/.db files”, we agree that such an agent defines a strictly more general task: the model can choose which files to open, write arbitrary Python, and call libraries such as pandas or SQL engines.
>
> Our goal with TQA-Bench is intentionally narrower and more controlled. The main benchmark configuration evaluates models without external tools, so that performance differences can be attributed to their intrinsic ability to understand and reason over long serialized tables, rather than to the design of a particular agent framework (tool set, retry policy, execution budget, etc.).
>
> To bridge this gap, **Appendix F** reports an additional study in which we keep the same questions and prompts but allow a frontier model (GPT-5.1) to call a code interpreter. We observe substantial gains on numerically heavy tasks such as correlation (e.g., COR accuracy improves from 53.54% to 60.53%), but performance is still far from perfect.
>
> This indicates that (i) TQA-Bench provides a challenging base layer even for coding agents, and (ii) in its primary configuration it offers a clean, tool-free benchmark specifically targeted at long-context multi-table QA.

---

> ### Comment · Reviewer_aAd1 · 2025-11-27
> **Reseponse to the authors**
>
> Thanks for your detailed response. However, your responses to my two major concerns are still not convincing to me.
> 1. On the benchmark contribution side, the argument that the benchmark provides “a controlled evaluation of long-context multi-table reasoning without external tools, which is not the focus of existing table QA or Text2SQL datasets” is unconvincing to me. E.g., Spider 2.0 can readily be evaluated under a prompting-only setting. I do not view this as establishing a necessary or substantially new contribution on the benchmark side.
> 2. On the method side, an agent-style baseline, even a simple openai-gym–like loop, would be appropriate and informative. As you noted, there is no consensus on agent frameworks, but such a baseline would allow the model to interact with the environment over multiple turns and self-correct, which aligns with the problem structure.
> Therefore, I will keep my score unchanged.

---

> > ### Author Response · Authors · 2025-12-02
> > **Question 1**
> >
> > Thank you again for the follow-up and for clearly summarizing your two remaining concerns. We understand that your score will remain unchanged, and we fully respect this. We would like to leave a short clarification of how we see the scope of our contribution and how recent additional experiments relate to your comments.
> >
> > > **Question 1:** On the benchmark contribution side, the argument that the benchmark provides “a controlled evaluation of long-context multi-table reasoning without external tools, which is not the focus of existing table QA or Text2SQL datasets” is unconvincing to me. E.g., Spider 2.0 can readily be evaluated under a prompting-only setting. I do not view this as establishing a necessary or substantially new contribution on the benchmark side.
> > >
> >
> > We agree that Spider 2.0 can, in principle, be evaluated under a prompting-only setting by serializing tables and providing them directly in the prompt. Our intention is therefore not to claim that “prompting-only long-context evaluation” is unique to TQA-Bench, nor that Spider 2.0 cannot be adapted in this way.
> >
> > Rather, TQA-Bench is designed to be complementary to Spider 2.0, with a more narrowly scoped but highly controlled focus:
> >
> > - We target fully materialized, long-context multi-table inputs (8K–64K tokens) where all relevant tables are serialized into the prompt and are fully observable to the model at once.
> > - Within this setting, we systematically vary serialization format, single- vs. multi-table structure, and symbolic extensions to probe how models scale with context length and table composition.
> > - We deliberately remove the external DB/tool dimension in the primary benchmark configuration so that performance differences can be attributed to the model’s intrinsic ability to parse and reason over long serialized tables, rather than to the design of a particular execution environment or Text2SQL pipeline.
> >
> > By contrast, Spider 2.0 is primarily constructed as an enterprise Text2SQL benchmark for agentic workflows over production-style databases, spanning multiple schemas, SQL dialects, and realistic tool chains. In this sense, Spider 2.0 and TQA-Bench occupy different points in the design space.

---

> > ### Author Response · Authors · 2025-12-02
> > **Question 2**
> >
> > > **Question 2:** On the method side, an agent-style baseline, even a simple openai-gym–like loop, would be appropriate and informative. As you noted, there is no consensus on agent frameworks, but such a baseline would allow the model to interact with the environment over multiple turns and self-correct, which aligns with the problem structure. Therefore, I will keep my score unchanged.
> > >
> >
> > We agree that agent-style baselines are important and align well with realistic usage. In the current submission, however, our primary focus is on the effect of long-context serialization itself. For this reason, all of our COR experiments (53.54% with direct prompting, 60.53% with a code interpreter) are conducted in a strict setting where all relevant tables are fully serialized into a single long textual context, and the code interpreter operates only over this text. This design isolates the question we care about most in this work: how well can models read and reason over long serialized multi-table inputs as context length grows.
> >
> > Following your suggestion, we also implement a **tool-using coding-agent baseline**: tables are converted into JSON files, uploaded, and accessed via tools in the GPT-5.1 API, allowing the model to interact with structured data through **multi-step tool calls**. In this richer setting, COR accuracy at the 64K scale increases to 92.86%, confirming that powerful coding agents with structured access can solve these tasks much more effectively.
> >
> > We see these two regimes as complementary: the paper systematically studies the strict, fully serialized-text regime where long-context reasoning itself is the bottleneck, while the file-based agent setup illustrates how TQA-Bench can also serve as a testbed for stronger agent workflows.

---

### Author Response · Authors · 2025-12-02
**Summary**

**Dear Area Chair,**

We post this brief author comment to summarize our contribution and rebuttal updates for your meta-review.

**Highlight of our contribution.** Across reviews, there is broad agreement that TQA-Bench tackles an important and timely problem: evaluating LLMs on **long-context multi-table QA** over realistic relational databases at different scales. Several reviewers consistently highlight three common strengths: (i) the benchmark and dataset design, including foreign-key–aware sampling, symbolic extension, and controlled context-length budgets; (ii) the breadth of evaluation over **28 open- and closed-source models**, multiple serialization formats, and both single- vs. multi-table and direct-prompting vs. Text2SQL settings; and (iii) the concrete empirical insights such as Markdown being the most effective serialization format and direct table prompting often outperforming Text2SQL in accuracy. Reviewer **aAd1** specifically commends the extensive benchmarking and focus on long-context behavior; Reviewer **6s7s** notes that the new multi-table dataset and released data could be useful for follow-up work; Reviewer **rikH** emphasizes the high-impact nature of large relational contexts and the diverse public data sources; and Reviewer **sY5W** highlights the scalable long-context sampling mechanism and symbolic extension strategy as key contributions. We are grateful for these positive assessments.

### Key Additional Experimental Results.

1. **Benchmark difficulty & relation to prior work.** We clarify that TQA-Bench is intended as a **controlled long-context layer** that is *complementary* to Spider 2.0, MMQA, and MTabVQA. Our primary setting fully serializes all relevant tables into a single 8K–64K-token prompt and removes external DB/tools to isolate end-to-end long-context reasoning. Under this strict regime, even GPT-5.1 at 64K context reaches only **53.54–60.53%** accuracy on the hardest analytical task (COR), indicating substantial remaining headroom.
2. **Agent-style baselines.** In the main experiments, both direct prompting and the code interpreter operate **only over the serialized text**, because our focus is on the effect of context length itself. Following Reviewer **aAd1**’s suggestion, we additionally ran a **tool-using coding-agent baseline**: tables are converted to JSON files and accessed via tools in the GPT-5.1 API. In this richer setup, COR accuracy at 64K rises to **92.86%**, confirming that powerful coding agents with structured access can perform much better and that TQA-Bench can also serve as a testbed for such agentic workflows.
3. **Additional analysis.** Following Reviewer **rikH**’s feedback, we add paired **McNemar tests**, **95% confidence intervals**, and additional qualitative error-case analysis. In response to Reviewer **sY5W**, we also provide more detailed dataset comparisons and join-depth statistics.

### Paper Presentation Updates

1. We more explicitly position TQA-Bench as **complementary** to Spider 2.0, MMQA, and MTabVQA, and we soften our wording on data contamination: using non-Wikipedia sources mitigates but does not provably eliminate leakage risk.
2. We clarify several construction and evaluation details that caused confusion, including: the exact counts of databases, templates, and question instances; how context-length budgets (8K/16K/32K/64K) are enforced across serialization formats; and the precise definition and grading of the COR and multiple-choice tasks.
3. We revise wording, captions, and examples in the main text and appendix to improve readability wherever reviewers indicated misunderstandings, while keeping the core problem setting and results unchanged.

These updates aim to strengthen the empirical evidence and clarify the presentation without altering the main scope of the work. We hope this concise summary, together with our detailed rebuttal and discussion, helps contextualize the reviews. Thank you very much for your time and consideration.

Best wishes,

Authors of Submission 7610

---

### Note · Authors · 2025-12-28

I have read and agree with the venue's withdrawal policy on behalf of myself and my co-authors.